# Gαq activation modulates autophagy by promoting mTORC1 signaling

Sofía Cabezudo [1,2,3,4,8], Maria Sanz-Flores [1,2,8], Alvaro Caballero [1,2], Inmaculada Tasset[5], Elena Rebollo[6], Antonio Diaz[5], Anna M. Aragay[7], Ana María Cuervo [5], Federico Mayor Jr[1,2,3✉] & Catalina Ribas [1,2,3✉]

The mTORC1 node plays a major role in autophagy modulation. We report a role of the ubiquitous Gαq subunit, a known transducer of plasma membrane G protein-coupled receptors signaling, as a core modulator of mTORC1 and autophagy. Cells lacking Gαq/11 display higher basal autophagy, enhanced autophagy induction upon different types of nutrient stress along with a decreased mTORC1 activation status. They are also unable to reactivate mTORC1 and thus inactivate ongoing autophagy upon nutrient recovery. Conversely, stimulation of Gαq/11 promotes sustained mTORC1 pathway activation and reversion of autophagy promoted by serum or amino acids removal. Gαq is present in autophagic compartments and lysosomes and is part of the mTORC1 multi-molecular complex, contributing to its assembly and activation via its nutrient status-sensitive interaction with p62, which displays features of a Gαq effector. Gαq emerges as a central regulator of the autophagy machinery required to maintain cellular homeostasis upon nutrient fluctuations.

[1] Departamento de Biología Molecular and Centro de Biología Molecular "Severo Ochoa" (UAM-CSIC), Madrid, Spain. [2] Instituto de Investigación Sanitaria La Princesa, Madrid, Spain. [3] CIBER de Enfermedades Cardiovasculares, ISCIII (CIBERCV), Madrid, Spain. [4] Structural Biology Program, Spanish National Cancer Research Centre (CNIO), Madrid, Spain. [5] Department of Developmental and Molecular Biology and Institute for Aging Research, Albert Einstein College of Medicine, Bronx, NY, USA. [6] Molecular Imaging Platform (MIP), Molecular Biology Institute of Barcelona (IBMB), Spanish National Research Council (CSIC), Barcelona, Spain. [7] Department of Biology, Molecular Biology Institute of Barcelona (IBMB), Spanish National Research Council (CSIC), Barcelona, Spain. [8] These authors contributed equally: Sofía Cabezudo, Maria Sanz-Flores. ✉email: fmayor@cbm.csic.es; cribas@cbm.csic.es

The highly conserved macroautophagy process, hereinafter referred to as autophagy, constitutes a core homeostatic process that contributes to proper cell function through the degradation and recycling of cytoplasmic constituents. Autophagy allows for lysosomal degradation of cellular components either in basal conditions or in response to perturbations of the intracellular or extracellular microenvironment[1,2]. Consistent with such a central role, alterations in autophagy-related pathways and machinery have been related to relevant pathological conditions, such as metabolic, degenerative, inflammatory, cardiovascular, or cancer diseases[3–5].

The molecular machinery involved in autophagy has been extensively characterized. In mammalian cells, membranes of varied origins (including endoplasmic reticulum, mitochondria, plasma membrane, or endosomes) form phagophores, which then expand and fuse to form autophagosomes, the transient double-membrane organelles that mediate cargo sequestration and delivery to lysosomes[6,7]. Lipidated LC3-I (referred to as LC3-II) is generated onto forming autophagosomes and allows for substrate loading upon binding to several receptors of autophagy, including p62/Sequestosome1 (p62/SQSTM1, hereafter p62). During autophagy, both LC3-II and the cargo receptors are degraded[8]. Autophagosomes fuse with lysosomes forming autolysosomes, where lysosomal hydrolytic enzymes degrade the engulfed substrates in order to release the resulting molecules to the cytosol. Finally, autolysosomes contribute to the regeneration of the lysosomal pool[9].

The signaling networks that underlie autophagy modulation in response to nutrient fluctuations, thus allowing cells to coordinate their metabolic activity and growth, remain to be fully understood. Changes in intracellular nutrient and energy status are mainly conveyed to the autophagic machinery via modulation of the mammalian target of rapamycin complex 1 (mTORC1) and AMP-activated protein kinase (AMPK) signaling cascades. In response to prolonged stress, autophagy is also regulated at the transcriptional level via transcription factors as TFEB or FoxO3[10].

The mTORC1 hub integrates signals from many stimuli, including amino acids, energy levels, glucose, oxygen, growth factors, and stress to coordinate the induction of anabolic processes[11] and the inhibition of catabolic ones, such as autophagy[12,13]. In response to growth factors, mTORC1 is mainly activated through PI3K/AKT pathway-mediated inhibition of the tuberous sclerosis complex 1/2 (TSC1/TSC2), thus allowing interaction of the lysosomal GTP-loaded Rheb with the mTOR catalytic domain and promoting its activation[11]. Amino-acid-induced activation of mTORC1 is mainly transduced through the Rag-GTPases, which mediate the translocation of mTORC1 from the cytoplasm to the lysosomal surface[14]. Therefore, full activation of mTORC1 on the surface of the lysosome is only achieved in the presence of both amino acids and growth factors[12]. Active mTORC1 represses autophagy by acting on both ULK1 and Vps34 complexes, therefore inhibiting autophagosome formation and maturation, and suppresses lysosome biogenesis through inhibition of TFEB. Conversely, different nutrient stress conditions inactivate the mTORC1 pathway at different levels, thus allowing triggering of autophagy, which enables cells to survive upon unfavorable conditions by providing nutrients, which in turn reactivate mTORC1 to terminate autophagy[12,13,15].

Therefore, the highly intertwined mTORC1-mediated modulation of autophagy upon nutrient imbalance requires a complex regulation to ensure cellular homeostasis. Although autophagy has been classically perceived as a cell-autonomous process, emerging evidence suggests that nutrients may also modulate this process in a systemic manner via different types of cell surface receptors. In particular, a variety of members of the G protein-coupled receptors (GPCRs) family have been proposed as nutrient sensors and suggested to be linked to the modulation of autophagy[16–18]. In our search for new effectors and cellular functions of the ubiquitous Gαq/11 subunits of heterotrimeric G proteins, which are coupled to many GPCR[19], we unveiled the key participation of Gαq in the modulation of autophagy in response to different types of nutrient stress and dissected the downstream signaling events responsible for this regulatory effect. Here, we demonstrate that Gαq/11 is central for the assembly of active mTORC1 multimolecular complexes and consequently for its inhibitory effect on autophagy.

## Results

**Gαq/11 is a potential modulator of autophagy**. An altered autophagic flux was noted when exploring potential differential phenotypic features in wild-type (WT) versus Gαq/11 knockout (KO) mouse embryonic fibroblasts (MEFs). Steady-state protein levels of the autophagy markers LC3-II and p62 were higher and lower, respectively, in Gαq/11 KO compared to WT MEFs in nutrient-rich conditions (10% serum). These differences were even more noticeable for LC3-II in low-serum conditions (Fig. 1a). The addition of lysosomal proteolysis inhibitors ($NH_4Cl$/Leupeptin) for different times to analyze autophagic flux revealed a markedly higher accumulation of LC3-II already at the earlier time point, consistent with an accelerated induction of autophagosome formation in Gαq/11 KO cells compared to WT cells in both nutrient contexts (Fig. 1b). A higher number of autophagic vacuoles (AV), mostly due to an increase in autolysosomes (AUT) in Gαq/11 KO than in WT cells was also noted in basal conditions using the tandem reporter mCherry-GFP-LC3 that allows tracking autophagic flux by the conversion of double fluorescence-labeled vesicles (autophagosomes) into only red fluorescence vesicles (autolysosomes) as GFP fluorescence is quenched in the acid lysosomal lumen (Supplementary Fig. 1a). Transmission electron microscopy (TEM) indicated an expansion of autophagic vacuoles in Gαq/11 KO compared to WT MEFs under nutrient sufficiency (Fig. 1c). Morphometry of TEM images revealed a significant increase in the cellular area occupied by autophagic vacuoles (Fig. 1d) at expenses of an increase in both number and size of autolysosomes in the Gαq/11 KO MEFS (Fig. 1e–f). In agreement with increased autophagic flux, we observed a higher ratio of autolysosomes to autophagosomes in cells lacking Gαq/11 (Fig. 1g and Supplementary Fig. 1a). Furthermore, quantification of these autophagic structures by flow cytometry also confirmed a significantly higher amount of autophagic vacuoles under basal or low serum conditions in cells lacking Gαq/11 (Supplementary Fig. 1b), which maintained intact its viability under these nutrient conditions (Supplementary Fig. 1c). Of note, stable re-expression of Gαq in the deficient background (Gαq/11 KO + Gq MEFs) restored the basal pattern of autophagy markers observed in WT MEFs (Fig. 1h), thus ruling out an unspecific autophagic phenotype of Gαq/11 KO MEFs unrelated to Gαq/11 dosage.

**Gαq localizes in lysosomal and autophagic compartments**. We examined if Gαq/11 was localized in autophagic compartments in physiological conditions, as described for different autophagy modulators[20]. Since it is difficult to detect endogenous Gαq protein with immunofluorescence techniques, we tested the presence of Gαq in autophagic/lysosomal compartments using subcellular fractionation of liver tissue and immunoblot. We uncovered the presence of Gαq in autophagosomes (APG) and autolysosomes (AUT) and, with less abundance, in lysosomes (Lys) isolated from fed rat livers (Fig. 1i). Upon a short period of starvation (4 h), reported to lead to the upregulation of macroautophagy[21], Gαq distribution increased in Lys, thereafter

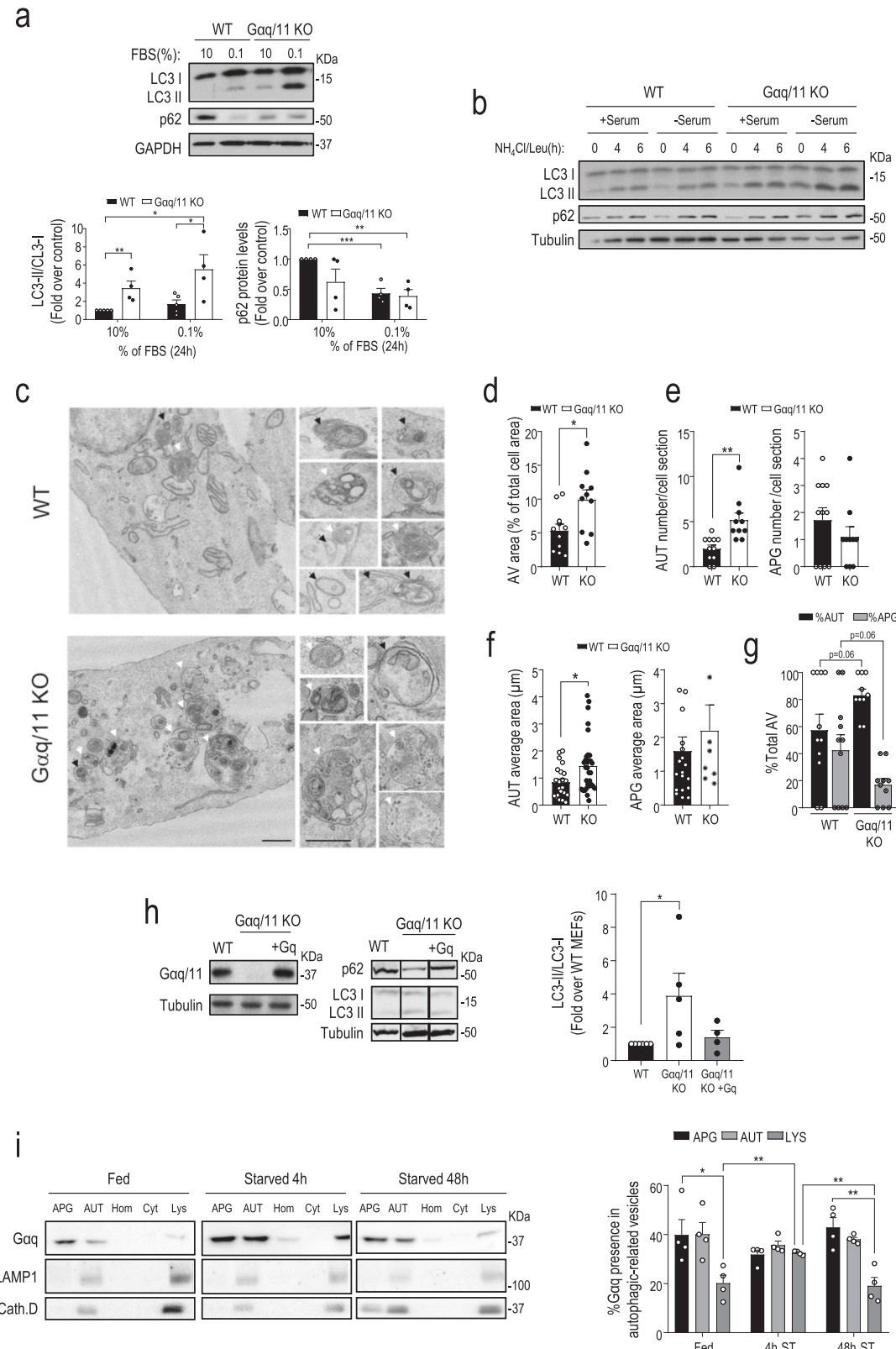

regaining its predominant location in APG/AUT after a longer starvation time (48 h) (Fig. 1i). Importantly, endogenous Gαq co-localized in these structures with key autophagy-related signaling components known to be located in such organelles as mTOR, Raptor, and p62 (Supplementary Fig. 1d).

Many components of the autophagic machinery and autophagy receptors (such as p62) present in these organelles are themselves substrates of the process being subjected to lysosomal degradation[20]. However, Gαq is not a substrate of autophagy under nutrient stress conditions, since serum removal did not promote a significant decrease of Gαq protein levels in WT MEFs, or in cells deficient for Atg5, a well-established model of macroautophagy deficiency, contrary to what was observed for the known autophagy substrate p62 (Supplementary Fig. 2).

**Fig. 1 Gαq/11 is a potential modulator of autophagy. a** WT ($n = 5$) and Gαq/11 KO ($n = 4$) MEFs were cultured for 24 h with 10% or 0.1% FBS and analyzed by immunoblot for LC3-I/II or p62 (WT $n = 4$, Gαq/11 KO $n = 4$) autophagic markers. Both markers were normalized with GAPDH. Data are mean ± SEM of the indicated independent experiments. LC3-II/LC3-I graph: WT (10%) vs. Gαq/11 KO (10%), P value = 0.0091; WT (0.1%) vs. Gαq/11 KO (0.1%), P value = 0.0389; WT (10%) vs. Gαq/11 KO (0.1%), P value = 0.0151. p62 graph: WT (10%) vs. WT (0.1%), P value = 0.0004; WT (10%) vs. Gαq/11 KO (0.1%), P value = 0.0012. **b** WT and Gαq/11 KO MEFs starved with serum-free DMEM for 16 h were treated with a combination of $NH_4Cl$ (20 mM) and leupeptin (100 µM) ($NH_4Cl$/Leu) during the last 4–6 h of the starvation period and autophagic markers assessed as above. A blot representative of three independent experiments is shown. **c** Representative transmission electron microscopy images of WT and Gαq/11 KO MEFs maintained at 10% FBS. Insets on the right show examples of autophagosomes and autolysosomes in WT and Gαq/11 KO MEFs. Scale bar, 0.5 µm. Black arrows indicate autophagosomes and white arrows autolysosomes. Representative images of three independent experiments are shown. **d–g** Morphometric analysis of micrographs as the ones shown in panel **c** to calculate the fraction of the cell occupied by autophagic vacuoles (AV) of WT ($n = 11$) or Gαq/11 KO ($n = 10$) MEFs, P value = 0.0133 (**d**), number of autolysosomes (AUT) and autophagosomes (APG) of WT ($n = 11$) or Gαq/11 KO ($n = 10$) MEFs, P value (AUT) = 0.0011 and P value (APG) = 0.3040 (**e**), the average area of autolysosomes (AUT), WT($n = 22$) and Gαq/11 KO ($n = 32$) MEFs, P value = 0.0146; and autophagosomes (APG), WT ($n = 19$) and Gαq/11 KO ($n = 8$) MEFs, P value = 0.4615 (**f**), the fraction of total AV contributed by AT and APG, WT ($n = 11$) or Gαq/11 KO ($n = 10$) MEFs, P values shown in the figure (**g**). Values are mean ± SEM of the indicated structures observed in >12 micrographs from three independent experiments. **h** Comparative analysis of autophagic markers (representative blot and graph) and Gαq/11 (representative blot) in WT ($n = 6$) or Gαq/11 KO ($n = 5$) MEFs and in cells re-expressing Gαq in the deficient background (Gαq/11 KO + Gq, $n = 4$) maintained in the presence of serum. Data are mean ± SEM of four independent experiments, P value = 0.0429. **i** Immunoblot analysis of Gαq and of the lysosomal/autolysosomal markers LAMP1 and Cathepsin D in autophagy-related organelles isolated from livers of fed or 4 and 48 h starved (ST) rats. Homogenate (Hom); cytosol (Cyt); autophagosomes (APG); autolysosomes (AUT); lysosomes (Lys). A representative blot is shown. Densitometry quantification values are expressed as the percentage of total Gαq protein associated to each autophagic/lysosomal compartment. Values are the mean ± SEM of four experiments. APG (Fed) vs. LYS (Fed), P value = 0.0316; LYS (Fed) vs. LYS (4 h ST), P value = 0.0091; LYS (4 h ST) vs. LYS (48 h ST), P value = 0.0081; APG (48 h ST) vs. LYS (48 h ST), P value = 0.0038. Statistical significance was analyzed using two-sided unpaired t-test. For all P values, *P < 0.05, **P < 0.005, ***P < 0.001. Source data are provided as a Source Data file.

Overall, the unexpected location of Gαq in autophagic compartments and lysosomes and the fact that this protein is not degraded under nutrient stress conditions in these organelles suggested that the observed changes in autophagy upon Gαq modulation may result from a regulatory function of this protein on autophagy.

**Cells lacking Gαq/11 display an earlier and prolonged autophagic response upon different types of nutrient stress along with an altered mTORC1 activation status**. Consistent with the accelerated autophagic flux observed in Gαq/11 KO versus WT MEFs, analysis of the temporal pattern of autophagy markers in these cells when subjected to low-serum conditions (Fig. 2a, upper blots), absence of amino acids (Fig. 2b, upper blots) or absence of glucose (Supplementary Fig. 3, upper blots), indicated that Gαq/11 KO cells displayed an earlier and sustained autophagic response in all nutrient stress contexts. p62 degradation and LC3-II accumulation initiated at <4 h in Gαq/11 KO cells (compared to 16 h in WT MEFs) following serum deprivation (Fig. 2a, upper blots), or at 5 min (compared to >30 min in WT MEFs) upon amino acids (Fig. 2b, upper blots) or glucose removal (Supplementary Fig. 3, upper blots). Moreover, these changes occurred in the same time frame that the increase in the formation of autophagic vacuoles detected by flow cytometry in Gαq/11 KO versus WT MEFs in all these experimental conditions (Supplementary Fig. 4). These data suggested that Gαq/11 plays a general and relevant role in the modulation of autophagy kinetics in different situations of nutrient stress.

The main cellular nutrient sensor pathways converge in the AMPK and mTORC1 cascades to regulate metabolism and autophagy. Under nutrient-rich conditions (10% serum), in agreement with their high basal autophagy, Gαq/11-deficient cells displayed a significantly lower basal mTORC1 activation status compared to WT cells in absence of changes in AMPK stimulation (Supplementary Fig. 5a), associated with lower proliferation rates (Supplementary Fig. 5b). Importantly, the earlier autophagic response observed in Gαq/11-deficient cells in the different nutrient stress contexts tested timely correlated with an earlier, more marked, and prolonged inactivation of the mTORC1 pathway compared to WT MEFs. The expected attenuation of the mTORC1 cascade in WT cells upon low serum (Fig. 2a), absence of amino acids (Fig. 2b), or glucose (Supplementary Fig. 3) was clearly enhanced and noted before in Gαq/11 KO MEFs, as assessed by the decreased phosphorylation status of the S6 ribosomal protein (p-S6), a central mTORC1 cascade downstream target. A similar reduced stimulation pattern was noted in Gαq/11-deficient cells for direct mTORC1 targets such p70-S6K or the phosphorylation of ULK1 in Ser-757 (Supplementary Fig. 6a, b), whereas AMPK pathway modulation was comparable in both cell types (Fig. 2a, b and Supplementary Fig. 3, lower blots) in all conditions tested. Overall, these results pointed out that Gαq/11 dosage might affect autophagy via the modulation of the mTORC1 signaling hub in a variety of nutrient stress contexts.

**The specific activation of Gαq/11 leads to the activation of mTORC1 and reverts the autophagic phenotype promoted by nutrient stress conditions**. We next sought to establish whether the activation of Gαq/11 was linked to the modulation of the mTORC1/autophagy pathways. Since canonical Gαq/11 stimulation takes place upon agonist binding to GPCR, we used the designer receptor exclusively activated by a designer drug (DREADD) coupled to Gαq/11 (DREADD-Gq system) stably expressed in HEK-293 cells. This model permitted to specifically trigger Gαq/11 activation by the synthetic ligand Clozapine-N-Oxide (CNO)[22] without interference from endogenous signals (scheme in Fig. 3a). As expected, CNO-mediated DREADD stimulation promoted rapid and transient ERK1/2 and Akt (Thr308) phosphorylation (positive controls of known Gαq downstream cascades), as well as clear mTORC1 pathway activation noted at slightly longer times of treatment (Supplementary Fig. 7a). Of note, in serum-starved cells (where downmodulation of mTORC1 pathway and upregulation of autophagy takes place (Fig. 3b, line 2) compared to conditions of 10% serum (Fig. 3b, line 1)) prolonged stimulation with CNO fostered a sustained activation of the mTORC1 cascade (as assessed by the p-S6 readout) eventually leading to inhibition of autophagy (decreased LC3-II and increased p62 levels) to return to levels comparable to the serum-supplemented conditions (Fig. 3b).

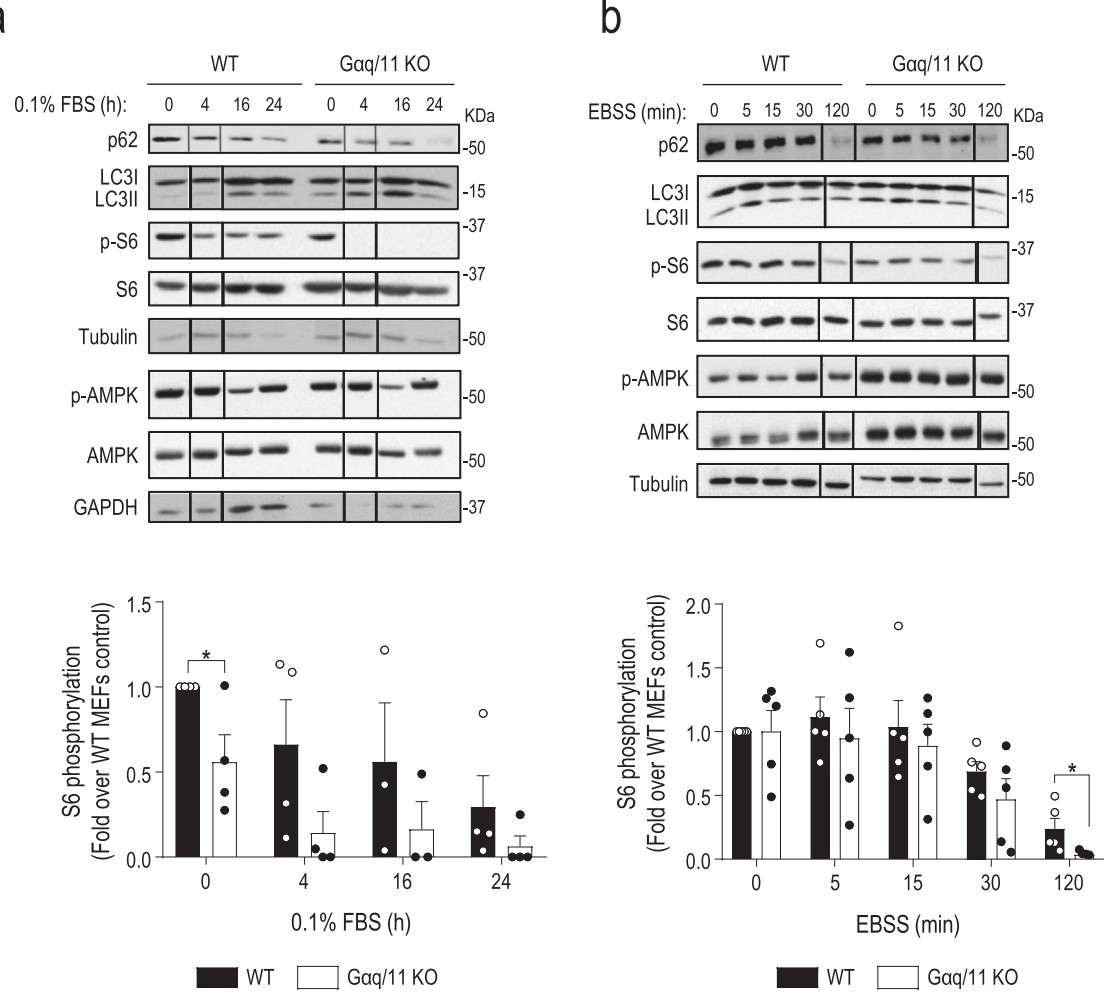

**Fig. 2 Cells lacking Gαq/11 display an earlier and prolonged autophagic response upon different types of nutrient stress along with an altered mTORC1 activation status. a** WT and Gαq/11 KO MEFs were maintained in presence or absence of serum during the indicated times ($n = 4$, except for time 16 h, $n = 3$). WT (0.1%, 0 h) vs. Gαq/11 KO (0.1%, 0 h), P value = 0.0341. A representative blot of four independent experiments is shown. **b** WT and Gαq/11 KO MEFs were maintained in the presence or absence of amino acids during the indicated times ($n = 5$). WT (EBSS, 120 min) vs. Gαq/11 KO (EBSS, 120 min), P value = 0.0407. A representative blot of five independent experiments is shown. For **a** and **b**, autophagic markers (LC3-II and p62) and the activation levels of the mTORC1 and AMPK pathways were analyzed by assessing the phosphorylation status of the downstream target of mTORC1 S6 ribosomal protein (see Supplementary Fig. 6a, b for additional readouts of this cascade) or of AMPK, respectively. Tubulin and GAPDH (for the p-AMPK and AMPK blots in panel **a**) were used as loading controls. Phospho-S6 data (mean ± SEM of the indicated independent experiments) were normalized using total S6 ribosomal protein. Statistical significance was analyzed using two-sided unpaired t-test. For all P values, *$P < 0.05$. Source data are provided as a Source Data file.

Thus, we subjected DREADD-Gq cells to either low serum or absence of amino acids nutrient stress conditions, and thereafter Gαq/11 was specifically stimulated with CNO for different times to check for potential mTORC1 reactivation and reversion of the induced autophagic phenotype (see the scheme of experimental approach in Fig. 3a). After 24 h in low-serum conditions (time 0 in the immunoblots shown in Fig. 3c), cells displayed the characteristic features of autophagy induction (inactivation of the mTORC1 pathway and increment of LC3-II levels compared to the 10% FBS control situation) (Fig. 3c). In control cells (without CNO stimulation), this pattern persists in the subsequent hours, as assessed by the decreased status of p-S6 (Fig. 3c) and pS757-ULK1 (Supplementary Fig. 7b) mTORC1 cascade readouts and high LC3-II levels (Fig. 3c) and only slowly reverts at 16–24 h, likely via the endogenous nutrients released by the autophagic process. Strikingly, the specific activation of Gαq/11 by CNO led to more marked and earlier (already observed at 1–4 h) reactivation of the mTORC1 pathway (Fig. 3c, p-S6 readout,

pS757-ULK1 readout Supplementary Fig. 7b), also leading to an earlier and stronger inhibition of autophagy, as detected by more rapidly decreasing levels of LC3-II at 1–4 h of CNO treatment compared to control non-Gq-stimulated cells (Fig. 3c). A similar pattern of autophagy modulation was detected when testing the evolution of p62 levels (Supplementary Fig. 7c). Once again, we did not detect changes in the AMPK pathway. It is worth noting that the reactivation of the mTORC1 pathway triggered by CNO does not completely parallel the pattern of Akt activation (Fig. 3c and Supplementary Fig. 7a), suggesting that other upstream mTORC1 modulators would be involved in the effects of Gαq/11. Consistent with this notion, a clear modulation of the mTORC1/autophagy cascade upon activation of the CNO/Gαq-cascade was observed in both 10 or 0.1% serum experimental conditions in the absence of parallel changes in p-S473 Akt phosphorylation status and also in the presence of Akti-1/2, a specific inhibitor of canonical PI3K-dependent Akt stimulation that prevents its phosphorylation by upstream kinases such as PDK1 (at T308)

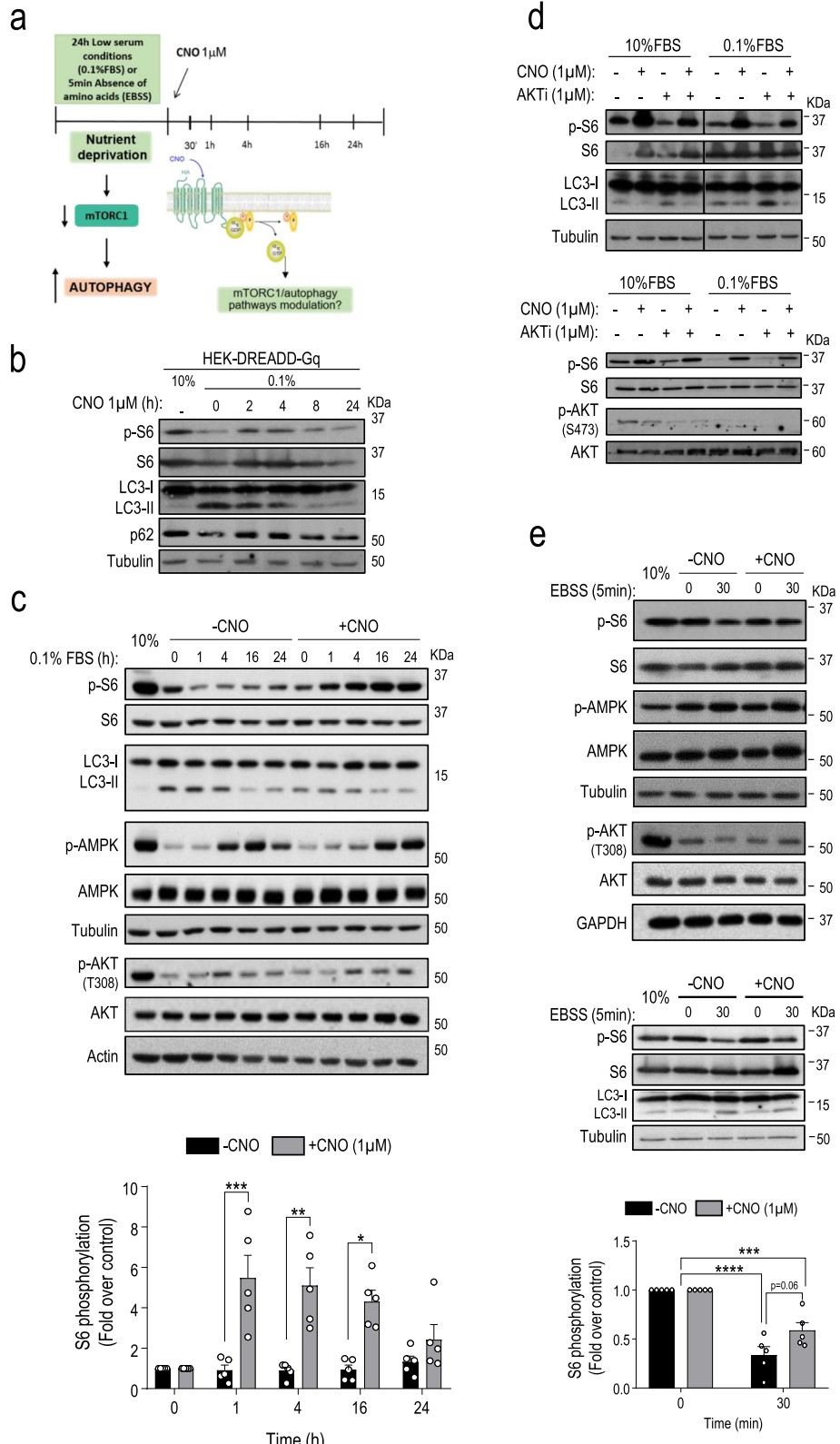

and mTORC2 (at S473)[23,24] (Fig. 3d and quantification in Supplementary Fig 7d), whereas it was markedly attenuated by the known mTOR kinase inhibitor rapamycin (Supplementary Fig. 7e). These data were consistent with the occurrence of alternative routes of Gq-mediated mTORC1 stimulation.

Similarly, stimulation with CNO for 30 min was also effective in preventing mTORC1 pathway decrease in more acute settings

such as short-term exposure of cells to amino acid-free EBSS media, as assessed by the status of p-S6 (Fig. 3e) or pS757-ULK1 and p-P70-S6K readouts (Supplementary Fig. 7f), or upon exposure of cells to amino acid-free RPMI media supplemented with dialyzed serum (Supplementary Fig. 7g), in the absence of changes in the Akt of AMPK pathways (Fig. 3e). CNO-mediated preservation of mTORC1 activation also attenuated the increase

**Fig. 3 The specific activation of Gαq/11 leads to the activation of mTORC1 and reverts the autophagic phenotype promoted by nutrient stress conditions. a** Experiment outline diagram. DREADD-Gq-HEK-293 cells growing in 10% FBS were starved with 0.1% FBS medium (24 h) or EBSS (5 min) and subsequently stimulated with vehicle or Clozapine-N-Oxide (CNO) (1 μM) for 1, 4, 16 or 24 h (serum context) or 30 min (amino acid context). **b** Cells growing in 10% FBS were starved for 16 h with serum-free medium and then stimulated or not with CNO (1 μM) for the indicated times. **c** Cells growing in 10% FBS were starved for 24 h with 0.1% FBS medium and stimulated with CNO or vehicle for different times. −CNO(1 h) vs. + CNO(1 h), P value = 0.0043; −CNO(4 h) vs. + CNO(4 h), P value = 0.0015; −CNO(16 h) vs. + CNO(16 h), P value = 0.0005. **d** Cells growing in 10% FBS or starved for 24 h with 0.1% FBS medium were stimulated with CNO (1 μM) or vehicle for 4 h in the absence or presence of the AKTi-1/2 inhibitor (1 μM). **e** Cells growing in 10% FBS were starved for 5 min with EBSS and stimulated or not with CNO for 30 min. ±CNO(0 min) vs. −CNO(30 min), P value <0.0001; ±CNO(0 min) vs. +CNO(30 min), P value = 0.008. In the different panels, autophagic markers (LC3-I/II, p62) were analyzed by western blot and the activation of mTORC1, Akt and AMPK pathways by assessing their phosphorylation status or that of downstream targets of mTORC1 (S6 ribosomal protein). Tubulin, actin, or GAPDH were used as loading controls. Blots are representative of three (panels **b**, **d**, and panel **e** lower blots) or five (panels **c** and **e**) independent experiments. Phospho-S6 data (mean ± SEM of five independent experiments) were normalized using total S6 ribosomal protein. See Supplementary Fig. 7b–d for additional readouts and graphs related to these experiments. Statistical significance was analyzed using two-sided unpaired *t*-test. For all P values, *P < 0.05, **P < 0.005, ***P < 0.001, ****P < 0.0001. Source data are provided as a Source Data file.

in LC3-II levels triggered by amino acid removal in control cells (Fig. 3e, lower panels and Supplementary Fig. 7h).

Taken together, these results support that, in different contexts of nutrient stress, specific activation of Gαq/11 triggers an earlier and marked attenuation of the autophagic process (as if the cells had partially recovered their nutrient levels) by promoting the activation of the mTORC1 pathway. Consistent with this notion, overexpression of constitutively active mutants of Gαq (Gq-R183C and Gq-Q209L) was able to maintain mTORC1 pathway activation even under low-serum conditions (Supplementary Fig. 7i).

**Gαq/11 is required to reactivate the mTORC1 pathway and thus inactivate autophagy in response to nutrient recovery.** We next explored whether Gαq/11 was required for the reactivation of the mTORC1 pathway that takes place once starved cells are exposed to physiological nutrient levels. This mTORC1 reactivation is essential to inactivate ongoing autophagy and return to basal conditions. Two different refeeding paradigms were used. In one set of experiments, WT and Gαq/11-deficient cells were starved for 24 h in low-serum conditions and then treated with increasing concentrations of serum for 18 h. Upon re-exposure to serum, WT MEFs were competent to trigger the mTORC1 pathway and to attenuate autophagy, while Gαq/11 KO MEFs were not able to fully reactivate the mTORC1 cascade, particularly at lower serum concentrations (Fig. 4a and Supplementary Fig. 8a, b). In another approach, cells were starved in an amino acid-free medium for 30 min followed by stimulation with a full amino acid mix for 5–60 min. Similarly, amino acid recovery induced a rapid mTORC1 pathway stimulation and consequent attenuation of autophagy in WT cells, but these nutrients were completely unable to promote mTORC1 cascade activation (as indicated by the status of p-S6, pS757-ULK1, or p-P70-S6K readouts) in Gαq/11 KO MEFs that maintained autophagy abnormally upregulated (Fig. 4b and Supplementary Fig. 8c, d).

As further evidence of Gαq/11-related changes in mTORC1-modulated autophagy, we found changes in intracellular positioning of endolysosomes in Gαq/11 KO cells compatible with basal repression of mTORC1 signaling and upregulation of autophagy. Immunofluorescence with LAMP1 to highlight the endolysosomal compartment revealed that WT MEFs displayed the expected switch of lysosomes from the cell periphery to a more perinuclear location upon exposure to amino acid-free medium[25], whereas the re-exposure to amino acids restored their original location (Fig. 4c). On the contrary, Gαq/11 KO cells showed a predominant perinuclear lysosomal distribution even in basal conditions, and such pattern was maintained in all the experimental conditions (Fig. 4c and Supplementary Fig. 9). These results are consistent with their inability to stimulate

mTORC1 in response to nutrient recovery. Of note, re-expression of Gαq in the deficient background (Gαq/11 KO + Gq MEFs) restored the pattern of lysosomal positioning observed in WT MEFs. Consistent with the involvement of activated Gαq/11 in mTORC1 modulation upon nutrient recovery, pretreatment of WT MEFs with the specific Gαq/11 inhibitor YM-254890 before amino acid supplementation markedly impaired its capacity to reactivate the mTORC1 pathway, as indicated by the status of p-S6, pS757-ULK1, or p-P70-S6K readouts (Fig. 4d). Overall, these data strongly suggested that Gαq/11 is a relevant upstream modulator of the highly intertwined mTORC1 complex. In order to confirm this notion in a physiological model, we treated freshly isolated liver explants from fed mice with YM-254890 or the well-known mTORC1 inhibitor Rapamycin for 1 h (scheme in Fig. 4e). The activated status of the mTORC1 pathway in fed conditions was inhibited to a similar extent by Rapamycin or the Gαq/11 inhibitor (Fig. 4f), which was also able to dampen canonical Gαq/11-mediated signaling pathways in such conditions (Supplementary Fig. 8e). Overall, our data indicated that Gαq/11 would help to link different types of nutrient signals with mTORC1 in order to modulate autophagy and coordinate metabolic responses to nutrient fluctuations.

**Gαq/11 is part of the mTOR/Raptor/p62 complex and favors its assembly in the presence of nutrients.** Intriguingly, immunoprecipitation of either endogenous Gαq (Fig. 5a) or mTOR (Fig. 5b) from WT MEFs under nutrient-sufficiency conditions revealed the presence of Gαq as part of the mTORC1 complex, together with other key components such as Raptor and p62. Importantly, immunoprecipitation of endogenous Raptor, a specific component of mTORC1, also revealed the presence in the same complex of Gαq, p62, and mTOR (Fig. 5c). Notably, cells lacking Gαq/11 displayed a lower association of p62 with mTOR (Fig. 5b) or Raptor (Fig. 5c), suggesting that Gαq could favor mTOR/p62/Raptor complex formation. The mTORC1 activation complex formation undergoes a dynamic modulation upon nutrient fluctuations[15,26]. Following amino acid deprivation, we found dissociation of p62 from mTOR, while Raptor and Gαq remained in the mTOR complex (Fig. 5d). In response to amino acid replenishment, p62 re-associated to the mTORC1 complex. These data suggested that Gαq/11 could favor the assembly of components of the mTOR/p62/Raptor complex in the presence of nutrients thus contributing to the activation of this pathway.

**p62 associates with Gαq through a PB1-like interaction and displays features of a Gαq effector.** We searched for potential molecular links between Gαq and mTORC1 activation. Interestingly, we found that expression of a Gq-Q209L/R256A/T257A

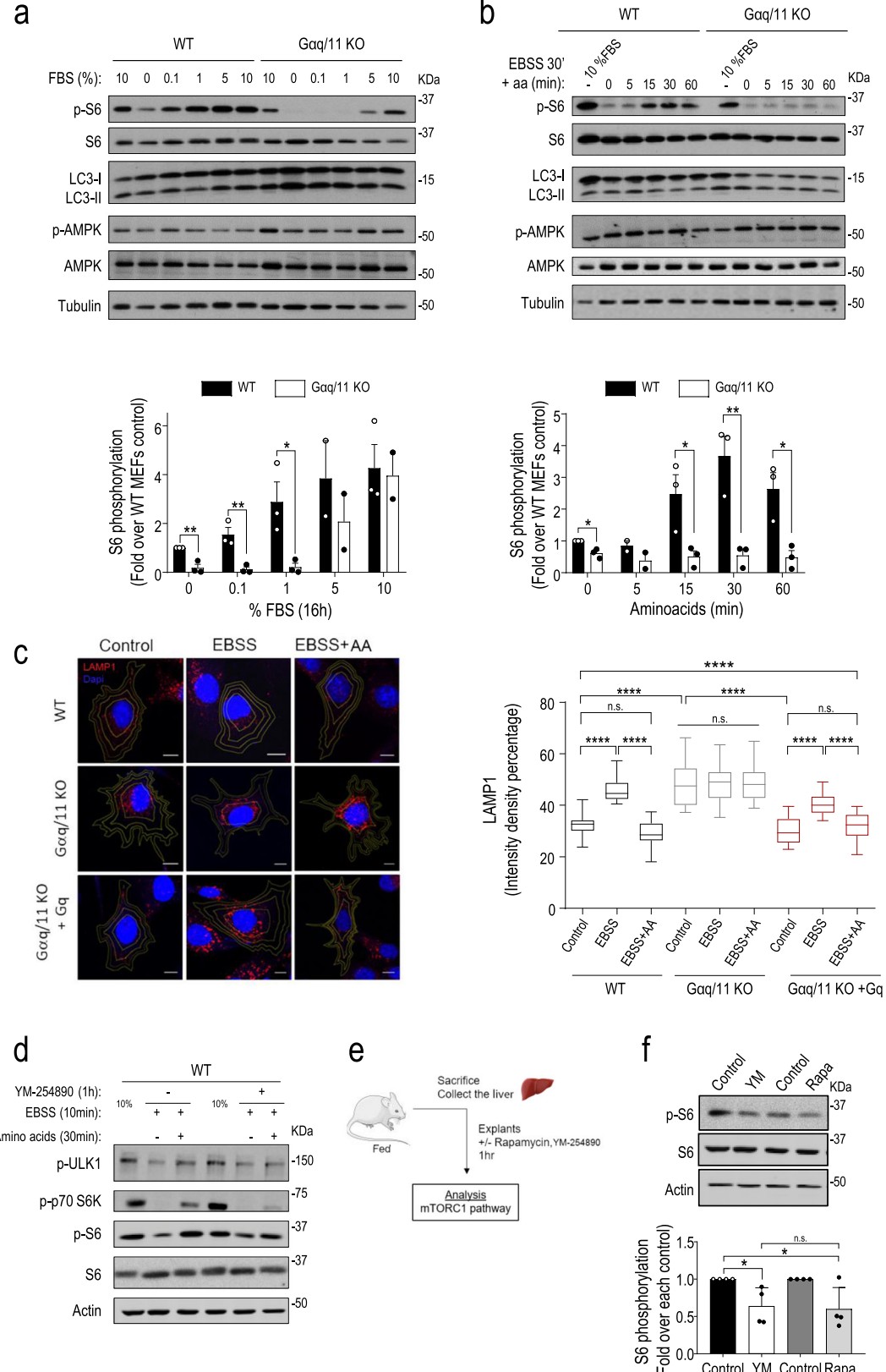

(GqQL-AA) constitutively active GTPase mutant that disrupts binding of Gαq to its canonical PLCβ and p63RhoGEF effectors[27,28], was able to promote activation of this pathway to a significant extent under low-serum conditions (Fig. 6a), suggesting that an additional effector binding through a different region was involved in Gαq-dependent mTORC1 modulation.

A novel acidic region in Gαq responsible for its direct interaction with the PB1-containing protein Protein kinase C ζ (PKCζ) has been reported, and glutamic acid residues 234 and 245 proven to be critical for such interaction[29]. Remarkably, the stimulation of the mTORC1 pathway triggered by transient expression of Gαq WT in low-serum conditions was not mimicked upon

**Fig. 4 Gαq/11 is required to reactivate the mTORC1 pathway and thus inactivate autophagy in response to nutrient recovery. a** WT and Gαq/11 KO MEFs were either maintained in the presence of 10% serum (control condition) or subjected to 24 h of starvation (0.1% FBS medium), followed by stimulation for 18 h with increasing doses of serum. WT vs. Gαq/11 KO MEFs $P$ value (0% FBS) = 0.0049; $P$ value (0.1% FBS) = 0.0099; $P$ value (1% FBS) = 0.0330. **b** WT and Gαq/11 KO MEFs were either maintained in the presence of 10% serum or subjected to a 30 min treatment with EBSS, followed by stimulation with a full amino acid mix for the indicated times. WT vs. Gαq/11 KO MEFs $P$ value (0 min) = 0.0112; $P$ value (15 min) = 0.0377; $P$ value (30 min) = 0.0090; $P$ value (60 min) = 0.0177. Autophagy markers (LC3-I/II) were analyzed by western blot (see quantifications in Supplementary Fig. 8a and c) and the activation of mTORC1 and AMPK pathways by assessing the phosphorylation status of downstream targets of mTORC1 (S6 ribosomal protein, see Supplementary Fig. 8b and d for additional readouts of this cascade) or AMPK, respectively. Tubulin was used as a loading control. Phospho-S6 data (mean ± SEM of three independent experiments) were normalized using total S6 ribosomal protein. Representative blots are shown. **a, b** Statistical significance was analyzed using two-sided unpaired $t$-test. For all $P$ values, *$P < 0.05$, **$P < 0.01$. **c** Altered intracellular positioning of lysosomes in Gαq/11 KO cells. Representative confocal micrographs of LAMP1 (red) and DAPI (blue) in WT, Gαq/11 KO and Gαq/11 KO + Gq MEFs in control conditions, after 30 min in amino acid-free medium (EBSS) and upon re-exposure to amino acids for 15 min (EBSS plus AA). Images are maximum intensity projections acquired using Zeiss LSM780 confocal microscope using a ×63 Oil (NA = 1.4) oil immersion lends. Scale bar, 10 μm. Image on the right shows the quantification analysis of lysosome distribution. The fluorescence intensity density per ring was measured and normalized as detailed in "Methods" and Supplementary Fig. 9. Control (Wt, $n = 30$; Gαq/11 KO, $n = 30$ and Gαq/11 KO + Gq, $n = 20$); EBSS (Wt, $n = 30$; Gαq/11 KO, $n = 30$ and Gαq/11 KO + Gq, $n = 20$); EBSS + aa (Wt, $n = 30$; Gαq/11 KO, $n = 28$ and Gαq/11 KO + Gq, $n = 22$). Data are mean ± SEM from the indicated cells for each condition from three independent different experiments. Statistical significance was analyzed using one-way ANOVA Sidak's multiple comparisons test. For all $P$ values, ****$P < 0.0001$. **d-f** The specific Gαq/11 inhibitor YM-254890 impairs reactivation of the mTORC1 pathway. **d** WT MEFs pretreated with YM-254890 (10 μM, 1 h) were maintained in the presence of 10% serum (control condition) or subjected to a 10 min treatment with EBSS, followed by stimulation with a full amino acid mix for 30 min and analysis of mTORC1 cascade readouts. Representative blots of three independent experiments are shown. **e** Diagram for mTORC1 pathway analysis in liver explants from fed mice subjected to YM-254890 (40 μM) or Rapamycin (500 nM) treatments for 1 h. The images used in this diagram are from Servier Medical Art (http://smart.servier.com/). **f** Normalized p-S6 data (mean ± SEM of four mice) and a representative blot are shown. Statistical significance was analyzed using two-sided unpaired $t$-test. For all $P$ values, *$P < 0.05$. Control vs. YM-254890 (YM), $P$ value = 0.0260; Control vs. Rapamycin (Rapa), $P$ value = 0.322. n.s. nonsignificant. Source data are provided as a Source Data file.

overexpression of the Gq-EEAA mutant (Fig. 6b). Overall, these results suggested that the PB1-binding region of Gαq is involved in mTORC1 pathway modulation.

Since the relevant autophagy receptor p62 contains both basic and acidic PB1 domains[30] and is present along with Gαq in the same mTORC1 multimolecular complex under nutrient sufficiency conditions (see above), we wondered whether the modulation of mTORC1/autophagy pathways could involve a PB1-mediated Gαq/p62 association. Interestingly, endogenous Gαq/p62 co-immunoprecipitation was detected in MEFs (Fig. 5a) and HUVEC (Supplementary Fig. 10a) cell types. We, therefore, used transient overexpression of Gαq and HA-p62 constructs to characterize their association. Of note, p62 displays features of a *bona fide* Gαq effector (reviewed in ref. [19]). Basal Gαq/p62 complex formation was fostered in a Gαq activation-dependent manner, being transiently enhanced (peaking at 15–30 min) upon Gαq activation by agonist-stimulated Gq-coupled M3 muscarinic receptors (Supplementary Fig. 10b), or in presence of constitutively active Gαq mutants (Fig. 6c). Consistent with data in Fig. 6a, b and with the notion that p62/Gαq association mediates mTORC1 modulation via noncanonical cascades, the GqQL-AA mutant, unable to interact with classical effectors as PLCβ and p63RhoGEF[27], was still able to efficiently bind p62 (Fig. 6d). Conversely, individual mutations to Alanine in conserved residues of the PB1-binding region of Gαq, E234, or E245 or a double E234/E245 mutation (Gαq-EEAA), markedly decreased association with p62 compared to wild-type Gαq, whereas the presence of the double mutant in a constitutively active Gαq background (Gαq-R183C-E234/E245-AA mutant, GαqRC-EEAA) was unable to potentiate the Gαq/p62 complex formation observed with wild-type Gαq (Fig. 6e and Supplementary Fig. 10c), confirming that E234 and E245 within the PB1-binding region of Gαq were also essential for the interaction of activated Gαq with p62.

In line with the features of a Gαq effector, Gαq/p62 association was disrupted by a well-established negative modulator of Gαq such as GRK2, but not by GRK2 mutants (GRK2-D110A) defective in Gαq binding (Supplementary Fig. 10d). Of note, Gαq mutants (Gαq-Y261F, Gαq-T257E, Gαq-W263D) unable to

bind GRK2 co-immunoprecipitated with p62 to an even greater extent than wild-type Gαq (Supplementary Fig. 10e), and were able to sustain mTORC1 pathway activation and modulate autophagy to a higher extent than wild-type Gαq under 24 h in low-serum conditions (Supplementary Fig. 10f), suggesting that the absence of Gαq competitors favors the interaction with p62 and mTORC1 stimulation via the Gαq PB1-domain binding region. Consistent with a PB1-like association between Gαq and p62, the overexpression of the PB1 domain of p62 clearly inhibited the complex formation (Fig. 6f). Of note, the PB1-mediated Gαq/p62 association displays determinants somewhat different from the Gαq/PKCζ interaction, since the Gαq-D236A mutant shows a slightly increased interaction with p62 compared to WT while decreased association with PKCζ (Supplementary Fig. 10c, g).

**mTORC1 and autophagy modulation by Gαq correlates with its ability to interact with p62.** Importantly, we found that the Gαq/p62 association was regulated by nutrient availability. A progressive dissociation of the Gαq/p62 complex was detectable at 16 h of serum starvation (Fig. 7a) and such decreased Gαq/p62 association was not due to potential autophagy-mediated p62 degradation in such long-term starvation conditions, since the same effect was observed in the presence of lysosomal inhibitors. Dissociation of the Gαq/p62 complex was also noted as earlier as 30 min of amino acids or glucose removal (Fig. 7b). Interestingly, such patterns of Gαq/p62 complex dissociation correlated in time with the deactivation of the mTORC1 pathway and the induction of autophagy previously observed in these different contexts of nutrient stress, suggesting a possible functional connection between these processes.

To further establish whether the modulation of mTORC1 and autophagy pathways by Gαq involved its interaction with p62, we performed reconstitution experiments in Gαq/11 KO MEFs and MEFs stably expressing, in a Gαq/11 KO background, similar levels of either Gαq wt (Gαq/11 KO + Gq MEFs) or Gαq mutant in its PB1-binding region with decreased Gαq/p62 interaction (+Gq-EEAA), respectively (Supplementary Fig. 11a). Basal activation status of the mTORC1 cascade and autophagy levels correlated with the ability of Gαq mutants to interact with p62.

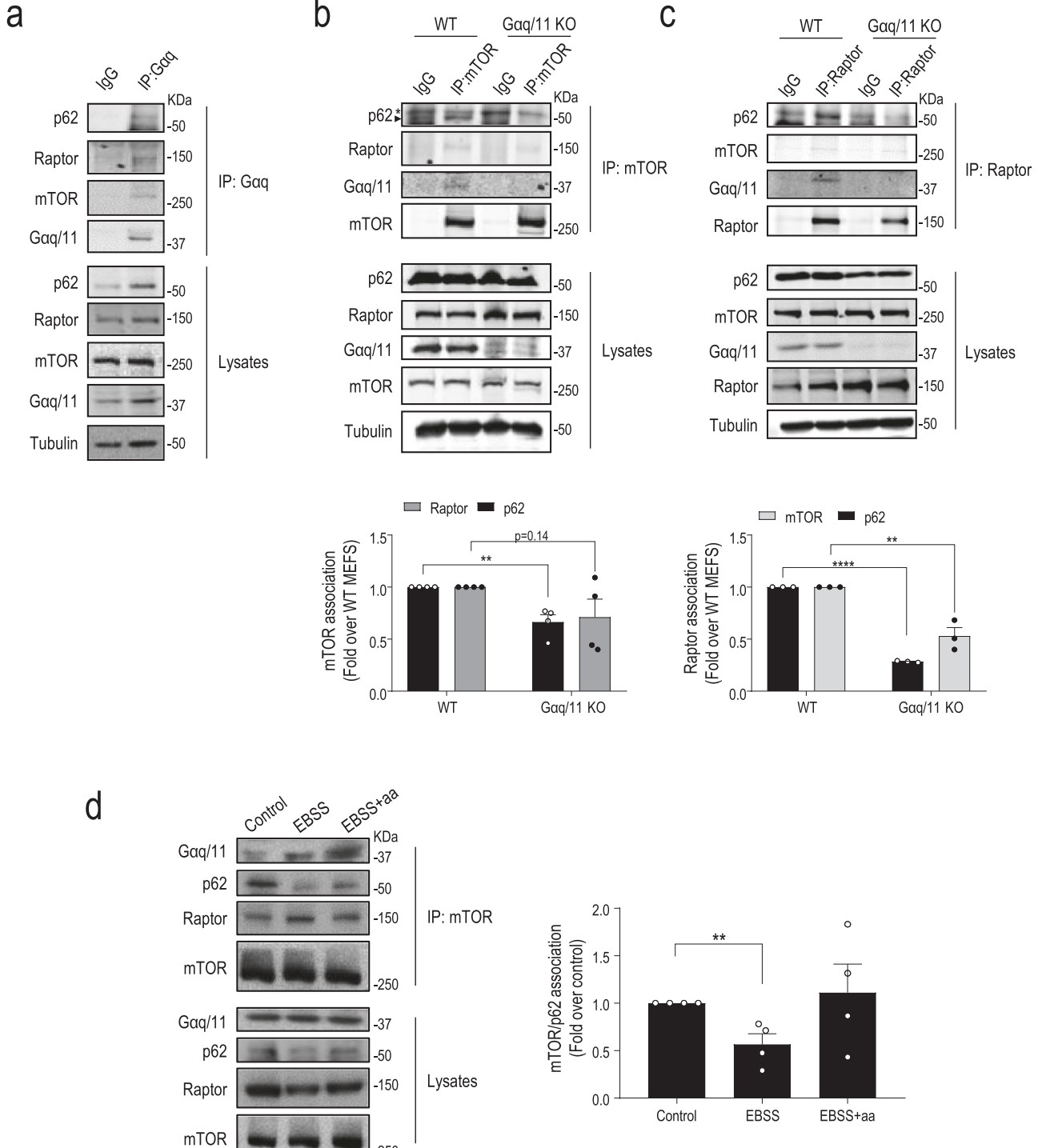

**Fig. 5 Gαq is part of the mTOR/p62/Raptor complex.** Immunoprecipitation of endogenous Gαq (**a**), mTOR (**b**), or Raptor (**c**) was performed in WT (**a**) or the indicated MEFs (**b**, **c**) under nutrient-sufficiency conditions (10% FBS). Blots are representative of three (**a**, **c**) and four (**b**) independent experiments. In **b** ($n = 4$) and **c** ($n = 3$), data (mean ± SEM of the indicated independent experiments) were normalized by total mTOR (**b**) or Raptor (**c**) and expressed as fold change of association with respect to the WT control condition. **d** Endogenous mTOR immunoprecipitation was performed as in panel **b** in WT MEFs treated with EBSS for 30 min and stimulated with a full amino acid mix for 15 min. Data (mean ± SEM of four independent experiments) were normalized by total mTOR and expressed as fold-change p62 association with mTOR with respect to the 10% FBS control condition. Statistical significance was analyzed using two-sided unpaired $t$-test. For all $P$ values, **$P < 0.01$, ****$P < 0.0001$. **b** WT vs Gαq/11 KO (p62/mTOR association), $P$ value $= 0.0030$. **c** WT vs. Gαq/11 KO (p62/Raptor association), $P$ value $< 0.0001$, and WT vs. Gαq/11 KO (mTOR/Raptor association), $P$ value $= 0.0047$. **d** Control vs. EBSS, $P$ value $= 0.0083$. Source data are provided as a Source Data file.

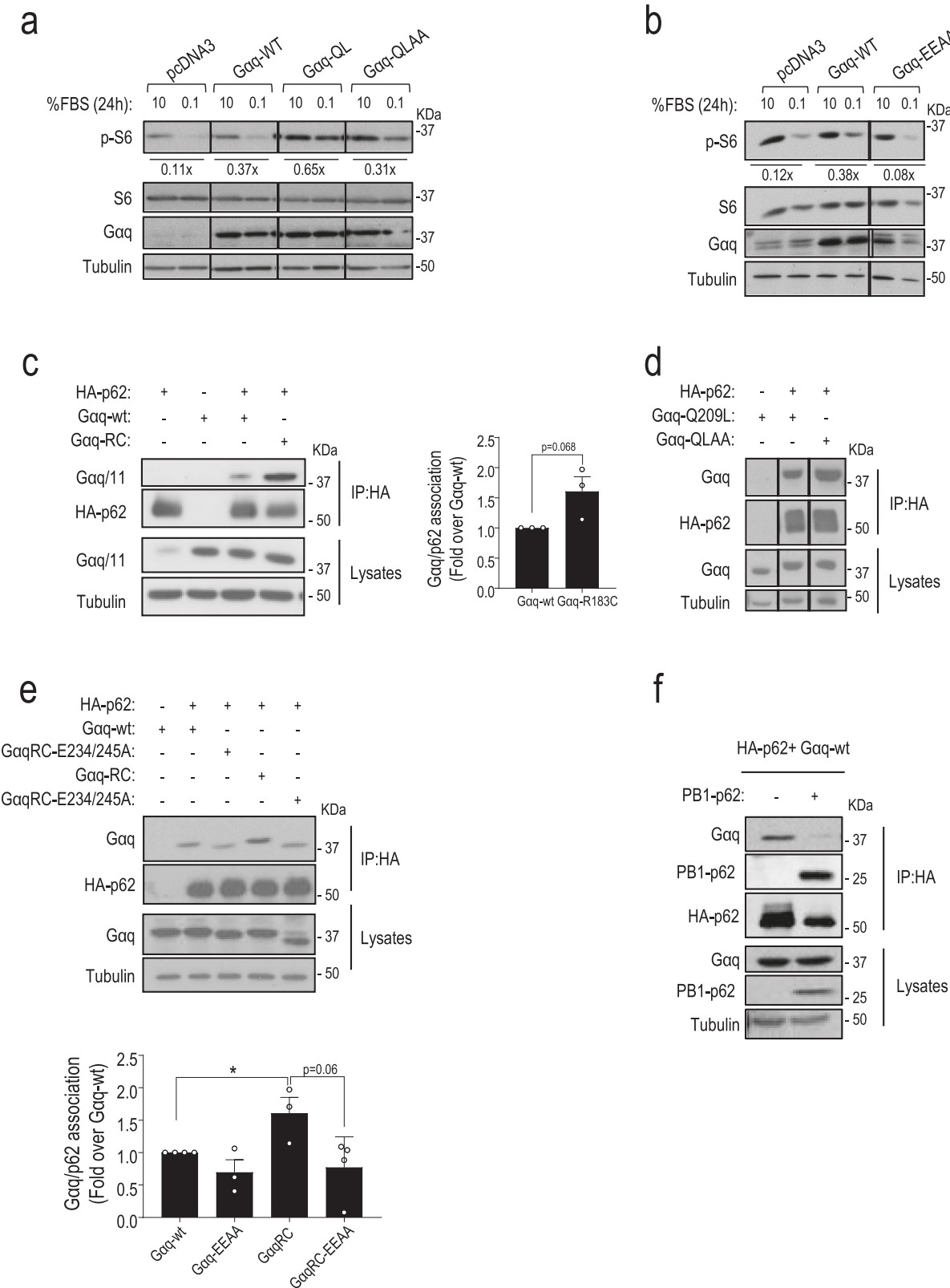

While Gαq/11 KO + Gq MEFs recovered the basal mTORC1 activation status of WT MEFs, cells displaying decreased Gαq/p62 interaction (+Gαq-EEAA MEFs) showed a pattern similar to parental Gαq/11 KO MEFs with significantly lower mTORC1 pathway activation and, consistently, enhanced autophagy, as indicated by higher LC3-II protein levels (Fig. 7c). Consistently, analysis of endogenous mTOR complexes showed a decreased association with Gαq and p62 in +Gαq-EEAA MEFs compared to Gαq/11 KO + Gq MEFs (Fig. 7d). In serum or amino acid starvation/recovery experiments, +Gαq-EEAA MEFs showed an inefficient pattern of reactivation of the mTORC1 pathway and maintained a high autophagy induction profile in response to amino acids or serum recovery, similar to the patterns detected in parental Gαq/11 KO MEFs (Fig. 7e and Supplementary Fig. 11b).

**Fig. 6 Gαq participates in the activation mechanism of mTORC1 via noncanonical effectors and associates with p62 through a PB1-like interaction.**
**a, b** DREADD-Gq HEK-293 (**a**) or CHO cells (**b**) transiently overexpressing Gαq wt or the indicated Gαq mutants (constitutively active Gq-Q209L, active but PLC-β activation-defective Gq-Q209L/R256A/T257A-GqQL-AA, mutant defective in binding to PB1-domain-containing proteins Gq-EEAA) were starved or not with 0.1% FBS for 24 h. The dephosphorylation fold of S6 upon serum deprivation was calculated with respect to the 10% FBS control of each cell population. A representative blot of three independent experiments is shown. **c–f** CHO cells were transiently transfected with the indicated combinations of HA-p62, Gαq WT or the indicated Gαq mutants (constitutively active Gq-Q209L or GqRC, active but PLC-β activation-defective GqQL-AA, defective in binding to PB1-domain-containing proteins Gq-EEAA and constitutively active GqRC-EEAA), or the PB1 domain of p62 (GFP-Flag-PB1-p62). In all cases (**c–f**), co-immunoprecipitation assays and expression controls in lysates were performed as described under "Methods". Representative blots of three independent experiments are shown. **e** Data (mean ± SEM of three (Gq-EEAA and GqRC) or four (Gαq WT and GqRC-EEAA) independent experiments) were normalized with total HA-p62 and expressed as fold induction of Gαq/p62 association over control. Statistical significance was analyzed using two-sided unpaired *t*-test. Gαq WT vs. GqRC, *P* value = 0.0312. Source data are provided as a Source Data file.

Conversely, Gαq/11 KO + Gq MEFs efficiently recovered the activation of mTORC1 and reversed the pattern of autophagy markers (accumulation of p62 and decreased LC3-II levels) in response to refeeding. Overall, these results support the notion that the interaction between Gαq and p62 in response to nutrients is key for the stimulation of the mTORC1 pathway and the control of autophagy.

## Discussion

We unveil here an unanticipated role of Gαq/11 as a key regulator of macroautophagy via modulation of the mTORC1 signaling hub. MEFs lacking Gαq/11 display higher basal autophagy and an earlier and prolonged autophagy induction upon serum removal or in the absence of amino acids or glucose than wild-type (WT) cells. Gαq/11-deficient cells are also unable to inactivate ongoing autophagy in response to nutrient replenishment. Our data suggest that Gαq acts as a general and relevant modulator of mTORC1 signaling over autophagy in response to fluctuations in different types of nutrients.

The metabolic modulation of autophagy is mainly orchestrated by the balance between the opposite inputs provided by the mTORC1 (negative) and AMPK (positive) pathways[12,31,32]. Our data provide insights into how Gαq/11 controls mTORC1 signaling and autophagy in response to the availability of different types of nutrients. First, cells lacking Gαq/11 display lower basal activation of the mTORC1 cascade and an earlier and sustained inactivation of this pathway compared to WT cells in response to all nutrient stress conditions tested (low serum, absence of amino acids, or absence of glucose), consistent with their enhanced autophagy response patterns. Gαq/11 dosage does not affect the modulation of the AMPK pathway under such nutrient stress conditions. Second, the specific stimulation of Gαq/11 via a synthetic DREADD-Gq-coupled GPCR promotes sustained activation of the mTORC1 pathway and triggers an earlier reversion of the autophagic phenotype promoted by serum or amino acids removal in parallel to the reactivation of the mTORC1 cascade (and independently of AMPK status). Moreover, constitutively active mutations in Gαq maintain mTORC1 activated in absence of serum. Third, while WT MEFs subjected to a period of starvation can fully reactivate the mTORC1 pathway and thus inhibit autophagy upon a rescue treatment with serum or amino acids, cells lacking Gαq/11 are unable to trigger such signaling response, therefore maintaining autophagy activated. The intracellular positioning of lysosomes in Gαq/11 KO cells is also compatible with the repression of mTORC1 signaling and upregulation of autophagy. Fourth, the Gαq/11 inhibitor YM-254890 prevents full reactivation of the mTORC1 pathway upon amino acid recovery in WT cells and inhibits the activated status of the mTORC1 pathway in fed mice-derived ex vivo liver explants to an extent similar to that attained in the presence of the well-known mTORC1 inhibitor Rapamycin. Fifth, consistent with its role as an autophagy modulator, we demonstrate the presence of an endogenous Gαq pool in autophagic compartments and lysosomes in rat liver, where it

co-localizes with mTOR, Raptor, and p62, and immunoprecipitation of endogenous mTOR or Raptor from WT MEFs in the presence of nutrients shows that Gαq is part of the mTORC1 multimolecular complex.

We postulate that Gαq participates as a general modulator of autophagy in response to serum, amino acids, or glucose by acting as a relevant component of the mTORC1 complex scaffolding machinery via interaction with the multifunctional p62/SQSTM1 protein. This signaling adaptor contains several functional regions and displays a variety of subcellular locations, including autophagosomes and lysosomes (reviewed in refs. [33,34]). p62 plays a central role as an autophagy receptor by forming PB1-linked homo-oligomers, recruiting ubiquitinated cargos, and inducing nucleation of the autophagosome membrane via its interaction with LC3[33] being itself a substrate of autophagy[35]. p62 is also involved in other cellular processes[33], including the modulation of mTORC1 (see below).

The mTORC1 signaling hub displays an intricate spatiotemporal regulation, involving both upstream signaling pathways and timely recruitment to lysosomes[12,13]. Major components of the mTORC1 complex include mTOR kinase, mLST8, DEPTOR, and the specific regulatory protein Raptor, which plays a relevant role in facilitating substrate recruitment and in controlling mTORC1 subcellular localization[12,13]. Many growth factors converge in promoting PI3K/Akt-mediated phosphorylation of TSC2 and in its subsequent dissociation from the lysosome, thus relieving its inhibition of the lysosomal Rheb GTPase, which becomes able to interact with mTORC1 and activate it[11,13]. Importantly, the presence of amino acids, acting through different sensors, is key for full mTORC1 activation by facilitating its recruitment to the lysosomal membrane. This is achieved via the Ras-related GTP-binding protein (Rag) family of small GTPases. An active Rag heterodimer (GTP-loaded RagA/B and GDP-loaded RagC/D) tethered by the Ragulator complex mediates the translocation of mTORC1 from the cytoplasm to the surface of lysosomes (see references in refs. [13,36–38]). Glucose-dependent mTORC1 stimulation is also partially regulated through Rag-GTPases by complex mechanisms (reviewed in refs. [31,39]). Thus, Rags appear to integrate multiple nutrient inputs converging in mTORC1 modulation. Importantly, in the presence of amino acids, p62 has been reported to interact with Raptor, thus helping mTORC1 complex translocation to the lysosome[40]. In addition, p62 is required for the interaction of mTORC1 with Rags, binds to RagC/D, and favors the formation of the active RagB-GTP/RagC-GDP heterodimer, which is further stabilized by Raptor[40]. The presence of p62 in the mTORC1 multimolecular complex might also contribute to the full activation of kinase activity via TRAF6-mediated K63 ubiquitination of mTOR[41].

A number of evidence support a role for p62 as an effector linking Gαq/11 to the activation status of mTORC1. First, p62, mTOR, Raptor, and Gαq are present in the same multimolecular

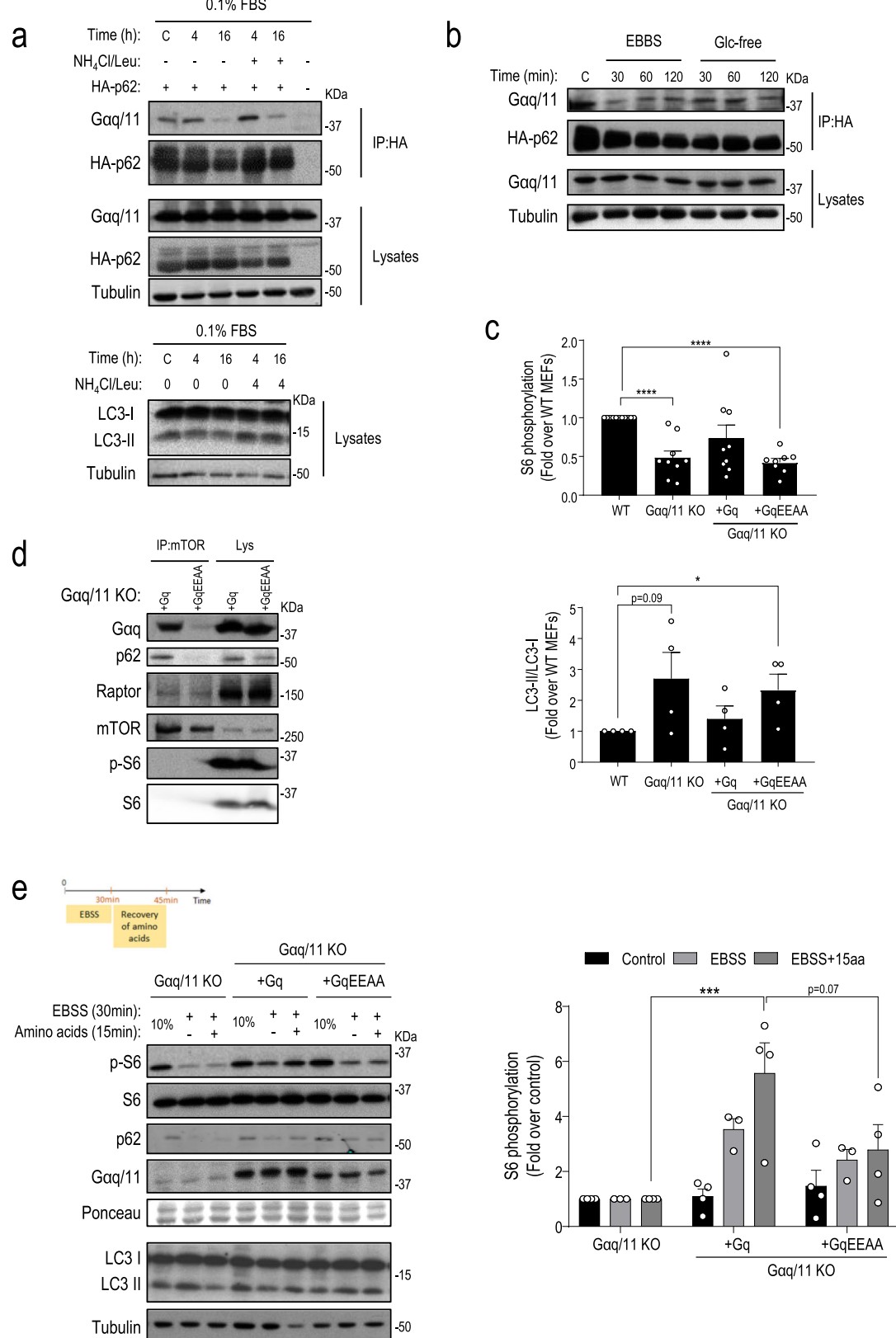

complex in endogenous conditions in the presence of nutrients, whereas cells lacking Gαq display a reduced presence of p62 in mTOR and Raptor immunoprecipitates along with decreased mTORC1 pathway activation. Second, p62 displays most of the features of a bona fide Gαq effector, including the association of endogenous proteins, enhanced interaction upon Gαq activation

and competition of Gαq/p62 interaction by other well-known Gαq interactors. Gαq and p62 appear to associate through a PB1-like interaction, involving the PB1 domain of p62 and the acidic PB1-binding region of Gαq, different from the specific interface involved in the association of Gαq with the canonical PLCβ or p63RhoGEF effectors[19]. This acidic Gαq region has been shown

**Fig. 7 Gαq/p62 association is regulated by nutrient availability and mTORC1 and autophagy modulation by Gαq correlate with its ability to interact with p62. a** CHO cells were transfected with HA-p62 and Gαq constructs, starved with 0.1% FBS and treated with a combination of NH₄Cl (20 mM) and leupeptin (100 μM) (NH₄Cl/Leu) (N/L) during the last 4 h of the starvation period, or with EBSS or glucose-free media for the indicated times (**b**). An LC3-I/II blot is shown to confirm the inhibition of autophagy by N/L. In both (**a**, **b**), co-immunoprecipitation assays and expression controls in lysates were performed as described under "Methods". Representative blots of three independent experiments are shown. **c** Basal activation status of the mTORC1 cascade and autophagy levels correlates with the ability of Gαq mutants to interact with p62. The basal activation status of the mTORC1 pathway in cells growing under nutrient-sufficiency conditions (10% FBS) was checked by analyzing the phosphorylation status of the S6 ribosomal protein in wild type MEFS, Gαq/11 KO MEFs and MEFs stably expressing in a Gαq/11 KO background either Gαq wt (+Gq MEFs) or a Gαq mutant (+Gq-EEAA) with decreased Gαq/p62 interaction. S6 phosphorylation data from immunoblot analysis (mean ± SEM of 11 (WT), 9 (Gαq/11 KO and Gαq/11 KO + Gq), and 8 (Gαq/11 KO + Gq-EEAA) independent experiments) were normalized using total S6 ribosomal protein and expressed as fold change with respect to the WT MEFs population. Basal autophagy levels were checked by analyzing LC3 protein levels by western blot and LC3-II/LC3-I ratios (mean ± SEM of four independent experiments) expressed as fold change with respect to the WT MEFs population. Statistical significance was analyzed using two-sided unpaired t-test. For all P values, *P < 0.05, ****P < 0.001. WT vs. Gαq11KO + Gq-EEAA, P value = 0.0402. **d** Endogenous mTOR immunoprecipitation was performed in Gαq/11 KO + Gq and Gαq/11 KO + Gq-EEAA expressing MEFs maintained under nutrient-sufficiency conditions (10% FBS). mTOR immunoprecipitates and total lysates were analyzed by western blot with specific antibodies to determine the components of the complex and overall cell expression levels, respectively. The blots shown are representative of three independent experiments. **e** mTORC1 reactivation and autophagy modulation in response to amino acid recovery after starvation correlate with the ability of Gαq to interact with p62. As shown in the experimental outline diagram, all cell lines were treated or not with EBBS for 30 min and then stimulated with a full amino acid mix for 15 min. mTORC1/autophagy pathways modulation were analyzed by western blot as in previous figures. S6 phosphorylation data (mean ± SEM of three (control and EBSS + 15 aa) and four (EBSS) independent experiments) were normalized using total S6 ribosomal protein and expressed as fold change over values in Gαq/11 KO MEFs population in each experimental condition. Statistical significance was analyzed using two-way ANOVA with Bonferroni multiple comparison test. Gαq/11 KO (EBSS + 15 aa) vs. Gαq/11KO + Gq (EBSS + 15 aa), P value = 0.0003. Source data are provided as a Source Data file.

to interact with the PB1 domain of PKCζ[29] or with the cold-activated channel TRPM8[42], consistent with this Gαq region being a functional module. Our data suggest that key Gαq acidic residues such as E234/E245 would engage in specific electrostatic interactions with canonical type II (positively charged) PB1 domains such as that present in p62. Importantly, the mTORC1 pathway activation status and the consequent reduction of autophagy correlates with the ability of Gαq mutants to interact with p62, as indicated by starvation/recovery experiments in Gαq/11 KO MEFs and MEFs stably expressing, in a Gαq/11 KO background, either WT Gαq (Gαq/11 KO + Gq MEFs), or a PB1-binding region Gαq mutant with decreased (+Gq-EEAA) Gαq/p62 interaction. Moreover, Gαq/p62 association is disrupted in response to diverse types of nutrient stress (serum removal, absence of amino acids or glucose) in parallel with mTORC1 cascade inactivation and consequent autophagy induction. On the contrary, recovery of amino acids after starvation restores co-immunoprecipitation of endogenous Gαq/p62/mTOR in WT MEFs, suggesting that the re-association of Gαq and p62 in response to nutrients facilitates the assembly of an active mTORC1 multimolecular complex.

Given the reported interactions of p62 (via regions different from its PB1 domain) with Raptor and Rags, leading to mTORC1 translocation to the lysosome and stabilization of active RagB-GTP/RagC-GDP heterodimers[40], it is tempting to hypothesize that nutrient-dependent interactions of Gαq with the PB1 domain of p62 may help to orchestrate a fully active mTORC/Raptor/p62/Rag complex. Such core Gαq scaffolding role would be consistent with its general role in autophagy control in response to different types of nutrient stress (see scheme in Fig. 8). In nutrient-sufficiency conditions, the presence of serum, amino acids, or glucose would foster the PB1-like interaction between Gαq and p62, leading to basal mTORC1 stimulation and preservation of low, homeostatic autophagy levels. Upon starvation, Gαq/p62 dissociation would facilitate the PB1-domain-dependent participation of p62 in the autophagic pathway[33]. The release of internal nutrients during autophagy (or the re-exposure to external fuels or serum) would in turn facilitate Gαq/p62/mTORC1 complex formation and reactivation in order to terminate autophagy. However, the contribution of other Gq-triggered cascades to Gαq-mediated modulation of mTORC1/autophagy pathways

cannot be ruled out. The canonical Gαq-governed PLCβ/IP3/Calcium/PKC pathways appear to be involved in the stimulation of mTORC1 by amino acids in pancreatic β cells[43] and C2C12 myoblasts[44], and calcium release from the endoplasmic reticulum can inactivate autophagy by maintaining mTORC1 activity[45]. Given the highly interconnected nature of the mTORC1 and Akt pathways, including the existence of complex feedback mechanisms[46], potential intermodulation of canonical Akt cascades and Gq/p62-mediated mTORC1 stimulation may also take place. Therefore, such pathways might also contribute to the overall effect observed, consistent with our findings showing that the modulation of the mTORC1 cascade exerted by Gαq/11 is only partially dependent on its canonical effectors.

Our results raise questions regarding the upstream signaling molecules triggering Gαq/11 activation in response to nutrients and the presence of Gαq/11 in lysosomal and autophagic organelles. An increasing number of nutrients and metabolites are being identified as endogenous ligands for Gq-coupled GPCR[16,17,47]. Intriguingly, a number of Gq-coupled GPCRs widely expressed in many cell types have been reported to sense amino acids and/or glucose, including the umami taste receptors (T1R1/T1R3), the calcium-sensing receptor (CaSR), GPRC6A, or metabotropic glutamate receptors[43,48,49]. T1R1/T1R3 receptors negatively modulate autophagy in response to amino acids through the activation of mTORC1 in a manner only partially dependent on PLCβ[50,51], whereas reduced T1R1/T1R3 expression in mice lacking T1R3 impairs mTORC1 stimulation and enhances autophagy in the absence of AMPK changes[43], a phenotype similar to that observed in Gαq/11 KO MEFs. On the other hand, relevant serum components as lysophosphatidic acid or sphingosine-1-phosphate and factors as insulin or IGF-1 have been reported to directly stimulate Gαq signaling[52].

The presence of Gαq in lysosomal and autophagy-related organelles may be subsequent to receptor-mediated activation at the plasma membrane or represent a stable pool able to undergo nutrient-mediated stimulation at such locations. Recent data indicate the presence of Gαq in endosomes and other organelles, such as mitochondria[49,53]. The agonist-dependent presence of GPCR and heterotrimeric G proteins in endosomes as a consequence of GPCR recycling or degradation mechanisms has also been reported[49]. Since endocytosis-derived organelles, can

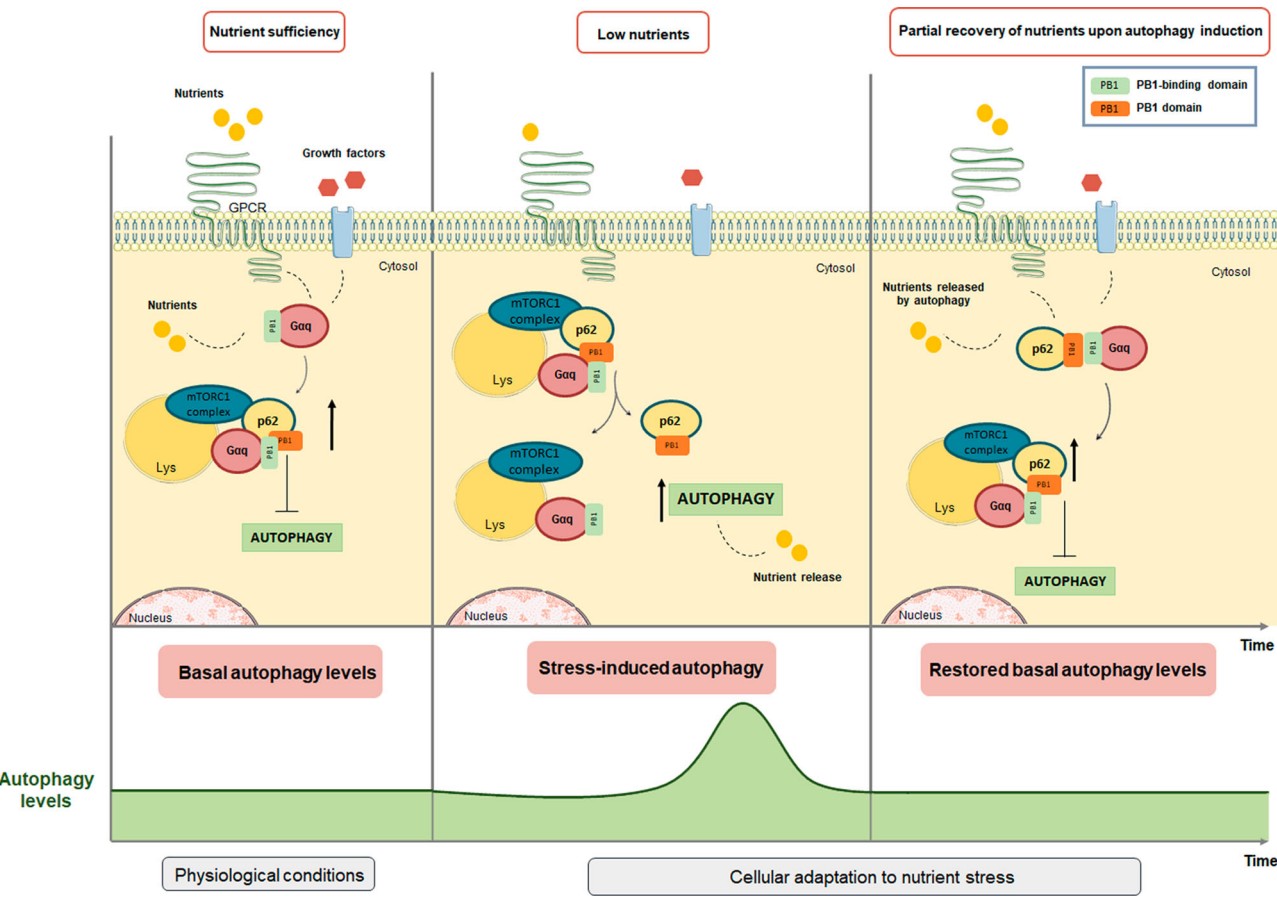

**Fig. 8 Proposed scaffolding role of Gαq in the assembly of active mTORC1 complex to control autophagy in response to different types of nutrient stress.** Upper part: In nutrient-sufficiency conditions, the presence of serum, amino acids, or glucose would foster the PB1-like interaction between Gαq and p62, leading to basal mTORC1 stimulation and preservation of homeostatic autophagy levels. Upon starvation, Gαq/p62 dissociation would facilitate the PB1-domain-dependent participation of p62 in the autophagic pathway. The release of internal nutrients during autophagy would in turn facilitate Gαq/p62/mTORC1 complex formation and reactivation in order to terminate autophagy. Lower part: Autophagy levels diagram. In response to starvation, there is a controlled and transient upregulation of autophagy, with a subsequent return to basal levels upon the partial recovery of nutrients.

form elongation membranes to finally form autophagosomes in response to starvation[54], endosomes might constitute a source for Gαq in autophagy-related organelles. Alternatively, activation of Gαq by nutrient-sensitive GPCR could take place directly at locations different from the plasma membrane via molecules that can gain access to the intracellular milieu via specific transporters, as is the case for amino acids, glucose, and other nutrients.

In sum, our results postulate Gαq as a key hub in the autophagy machinery. The increasing links of altered autophagy with relevant pathological conditions, such as metabolic disorders, cancer, or cardiovascular diseases[3–5], along with the central role of Gαq/11 and Gq-coupled GPCR in the same contexts[55,56], suggest that the function or malfunction of this Gαq/mTORC1/autophagy modulation cascade may contribute to these pathological situations and constitute a potential therapeutic target.

## Methods
### Experimental model and subject details
*Rats.* Adult male Wistar rats (200–250 g/8 weeks) fed or starved (4–48 h) were used under an institutionally approved animal study protocol. All rats were housed under pathogen-free conditions and handled according to protocols approved by the Institutional Animal Care and Use Committee of Albert Einstein College of Medicine (New York, USA).

*Mice.* Adult (8–12 weeks) male inducible endothelium-specific Gαq/Gα11-deficient mice (Tie2-CreERT2; Gnaq f/f; Gna11 − /− [EC-q/11-KO][57,58]) fed with a standard diet (SD, providing 13% of total calories from fat, 67% from carbohydrate and 20% from protein; 2014S Rodent Maintenance Diet, Teklad, Harlan, IN, USA) were used. Since for these experiments mice were not treated with tamoxifen, we refer to them as WT mice as they display endogenous Gαq levels. Mice were housed under a 12-h light–dark cycle with free access to food and water and under specific pathogen-free conditions at 20–24 °C with a humidity of 55% ± 10%. All animal experimentation procedures conformed to the European Guidelines for the Care and Use of Laboratory Animals (Directive 86/609) and were approved by the Ethical Committees for Animal Experimentation of our University and the Comunidad Autónoma de Madrid, Spain. We did not use randomization in our animal studies. We were not blinded to the group in our animal studies.

*Cell lines.* Gαq/11 WT and Gαq/11 KO MEFs were a kind gift from Dr. S. Offermanns (Max-Planck-Institute for Heart and Lung Research, Germany). Atg5 WT and Atg5 KO MEFs were provided by Dr. N. Mizushima (University of Tokyo, Tokyo, Japan) and DREADD-Gq-HEK-293 cells (human female) by Dr. Silvio Gutkind (University of San Diego, California, USA)[22]. CHO cells overexpressing the muscarinic M3 acetylcholine receptor (CHO-M3) (mice female) were a kind gift from Dr. A. B. Tobin (University of Glasgow, UK). Cell lines were authenticated by short tandem repeat (STR) or DNA barcoding analysis and tested for mycoplasma contamination. MEFs and DREADD-Gq cells were maintained in Dulbecco's modified Eagle's medium (DMEM) (Gibco Life Technologies), CHO-M3 cells in αMEM (Gibco Life Technologies), supplemented with 10% (v/v) fetal bovine serum (Sigma-Aldrich) and 1 mM L-glutamine, 50 IU/ml penicillin, 50 μg/ml streptomycin. Human umbilical vein endothelial cells (HUVEC) (human male) were grown in 199 Medium (Life Technologies, without NaHCO₃) supplemented with 2.2 g/L NaHCO₃ (pH 7.4), 1 mM L-glutamine, 50 IU/ml penicillin, 50 μg/ml streptomycin, and 15% FBS. After the first passage, cells were supplemented with 50 μg/ml endothelial cell growth supplements (Sigma-Aldrich) and 100 μg/ml heparin (Sigma-Aldrich). All cell lines were maintained at 37 °C in a humidified 5% CO₂ atmosphere.

## Method details

*Ex vivo liver explants.* Freshly collected liver explants from fed male mice were placed in dishes with DMEM in the presence or absence of the Gαq/11/14 inhibitor (YM-254890, 40 μM), rapamycin (500 nM), or histamine (200 μM) and then transferred to a 37 °C incubator at 5% $CO_2$ for 1 h[59]. Explants were homogenized using metal beads in a Tissue Lyser (Qiagen, Hilden, Germany)[60], and lysates used for immunoblot analysis of mTORC1 pathway modulation.

*Cell treatments.* For autophagy modulation by nutrient depletion assays, cells were maintained in 0.1% (v/v) bovine serum, Earl's balanced salt solution (EBSS (1 g/L D-glucose 2.2 g/L NaHCO3, 0.4 g/L KCl, 6.8 g/L NaCl, 0.6 g/L $H_2NaPO_4 \times 2H_2O$, 0.26 g/L $CaCl_2 \times 2H_2O$, 0.2 g/L $MgSO_4 \times 7H_2O$) or RPMI 1640 medium without amino acids, sodium phosphate (US Biological R8999-04A) supplemented with dialyzed serum during the indicated time periods. Nutrient recovery experiments in MEF cells were carried out by adding increasing percentages of bovine serum or a full amino acid mix (MEM amino acids solution (50×)) (Sigma-Aldrich). For autophagic flux assays, cells were treated with chloroquine (50 μM, Sigma-Aldrich), and a combination of the lysosomal proteolysis inhibitors ammonium chloride (20 mM, Sigma-Aldrich) and leupeptin (100 μM, Sigma-Aldrich), during the indicated times. For degradation assays, cells were treated with the lysosomal inhibitors mentioned and the proteasomal inhibitor MG132 (0.1 μM, Sigma-Aldrich) during the indicated time. DREADD-Gq HEK-293 cells were stimulated with Clozapine-N-Oxide (CNO) (1 μM, Tocris) during different times after being serum- or amino acids-starved. Gαq/11 inhibition was carried out by cell treatment with the specific Gαq/11/14 inhibitor (YM-254890, 10 μM, Wako) in serum-free media for 1 h. CHO-M3 cells were stimulated with carbachol (10 μM, Sigma-Aldrich) in serum-free media, during the indicated time periods. In order to analyze the implication of mTORC1 or PI3K/AKT cascades in Gαq modulation of the mTORC1 pathway, DREADD-Gq HEK-293 cells were treated with mTOR (Rapamycin, 500 nM, Calbiochem) or AKT (AKTi-1/2, 1 μM, Tocris; Ref: 5773) inhibitors.

*Plasmid DNA and transient transfections.* pcDNA3 was from Invitrogen and Gαq-Q209L cDNA from Missouri S&T cDNA Resource Center. The Gαq and Gαq-R183C cDNA were from our laboratories (Dr. A. M. Aragay, CSIC, Barcelona, Spain). The constitutively active Gαq mutant protein that lacks the ability to interact with PLCβ (Gαq-Q209L/R256A/T257A) was provided by Dr. R. Lin (Stony Brook University, New York, USA). The Gαq mutant proteins that lack the ability to interact with GRK2 (Gαq-T257E, W263D, Y261F) were a gift from Dr. T. Kozasa (The University of Tokyo, Japan). The PB1-binding region Gαq mutants (Gαq E234A, D236A, D243A, E245A, E234/E245-AA, Gq-R183C-E234/E245-AA) were generated in our laboratory using the QuickChange® site-directed mutagenesis kit (Stratagene) following the manufacturer's instructions[29]. GRK2, GRK2-K220R, GRK2-D110A cDNAs were a gift from Dr. J. L. Benovic (The Kimmel Cancer Center, USA). The GRK2-RH domain cDNA[61] was generated in our laboratory. The HA-p62 construct was provided by Dr. E. Campbell (University of Illinois at Chicago, USA). GFP-Flag-PB1-p62 was provided by Dr. B. Berk (University of Rochester, NY, USA). mCherry-GFP-LC3 (Addgene-84572, Dr. N. Mizushima, constructed by T. Kaizuka (University of Tokyo, Japan)). cDNAs for lentiviral construction pMD2.G, pSD.44 and psPAX2 (Addgene-12259, 12252, 12260, respectively) by Dr. R. Iggo (University of St Andrews, Edinburgh, UK). The desired combinations of cDNA constructs were transfected using the Lipofectamine 2000 method (Life Technologies), following the manufacturer's instructions.

*Cloning and mutagenesis.* psd44.Gq-wt and psd44.Gq-E234-E245-AA were created in our laboratory from psd44.GqQL (Addgene-46826, Dr. A. Mariotti) using the QuickChange Lightning Site-Directed Mutagenesis Kit, following the manufacturer's instructions (Agilent Technologies, 210,518). The constitutive activation mutation was reverted with the following PCR primers: GqQ209L_FW: GAT GTA GGG GGC CTA AGG TCA GAG AGA and GqQ209L_REV: TCT CTC TGA CCT TAG GCC CCC TAC ATC. Subsequently, Gαq PB1-binding region mutations were performed with the following primers: GqE234A_FW: CTAGTAGCGCTT AGTGCATATGATCAAGTTCTCGTGG and GqE234A_REV: CCACGAGAA CTTGATCATATGCACTAAGCGCTACTAG, GqD236A_FW: GTA GCG CTT AGT GAA TAT GCG CAA GTT CTC GTG GAG TCA, GqD236A_REV: TGA CTC CAC GAG AAC TTG CGC ATA TTC ACT AAG CGC TAC (Sigma-Aldrich). All primers' information has been included in Supplementary Table 1.

*Generation of stable cell lines.* For the generation of MEFs stably expressing WT or Gαq PB1-binding region mutants in a Gαq/11 KO background, firstly psd44.Gq-wt and psd44.Gq-E234-E245-AA DNA constructs were transfected together with psPAX2 and pMD2.G in HEK-293-T packing cells, using Metafectene (Bointex) according to the manufacturer's instructions, for the generation of the lentiviral particles. Lentiviral purification and Gαq/11 KO MEFs transduction were performed[62]. All generated MEF cells lines, psd.44.Gq-wt (Gαq/11 KO-rGq MEFs) and pds44.Gq-EEAA (rGq-EEAA MEFs) MEFs were maintained under 1 μg/ml puromycin treatment.

*Generation of mCherry-GFP-LC3 MEFs stable cell lines.* mCherry.GFP-LC3 stable overexpressing cells were generated in Gαq/11 WT and Gαq/11 KO MEFs. Cells were seeded in a six-well plate a day before transduction. Then, a 2× polybrene

mixture was prepared by adding 0.5 ml medium + 1μl polybrene stock solution (10 mg/ml) to the cells together with 0.5 ml mCherry-GFP-LC3 virus supernatant generated by the same protocol as for psd.44 lentivirus vectors. Next, 24 h after transduction, 1 ml of regular medium was added to the cells without removing the virus supernatant. After 48–72 h, cells were checked for positive GFP signal by fluorescence microscopy and seeded for the experiment. Depending on the cell type these stable cell lines were used until passage 3–4.

*Immunoprecipitation assays.* Cells were lysed with the low detergent lysis PD buffer (50 mM Tris, pH 7.5, 150 mM NaCl, 1 mM NP-40, 0.25% (w/v) sodium deoxycholate, 1 mM EGTA pH 8.0, 1 mM NaF, protease inhibitors) 24–48 h after transfection and clarified by centrifugation. Immunoprecipitation was performed with agarose-conjugated anti-HA antibodies (Santa Cruz, F-7, # sc-7392-AC). Endogenous immunoprecipitation assays were performed by incubating 0.5 μg/ml of p62 antibody (GP62-G, Progen), mTOR antibody (7C10, Cell Signaling), Raptor (24C12, Cell Signaling), or Gαq antibody (E-17, Santa Cruz) or a control IgG (normal rabbit IgG sc-2027, Santa Cruz) with 5 mg of cell lysates in the presence of BSA (1 mg/ml), followed by re-incubation with Recombinant Protein G-Sepharose™ 4B (Invitrogen, 101243).

*SDS-PAGE and immunoblot analysis.* The PD buffer supplemented with protease inhibitors and 1 mM PMSF was used to make cell lysates. Protein concentration was measured using the DC Protein Assay Reagents A, B, and S (Bio-Rad). Protein lysates were separated on 8–15% SDS-PAGE gels, transferred onto the nitrocellulose membrane (Bio-Rad) and blocked for 1 h in 5% BSA–TBS. After incubation with primary antibodies (overnight) and secondary antibodies (1 h), the membranes were washed and analyzed using ECL (enhanced chemiLuminiscence) from Amersham Pharmacia Biotech and AGFA films or the LI-COR Odyssey Infra-red Imaging System. Immunoreactivity bands were quantified by laser densitometry with a Biorad GS-900 scanner and using de Bio-Rad provided Image Lab 5.2 or by the software included in the Odyssey Infra-red Imaging System. Vertical lines in western blots indicate juxtaposed lanes that come from the same gel but were nonadjacent. Primary antibodies used in western blotting with dilutions were as follows: AKT (Cell signaling #9272; 1:1000), p-AKT (S473) (Cell Signaling #4060; 1:1000), p-AKT (T308) (Cell Signaling #9275; 1:1000), AMPKα (Cell Signaling #2532; 1:1000), p-AMPK (T172) (Cell signaling #2535; 1:1000), Tubulin (Santa Cruz #SC-53030; 1:1000), Erk1 (C-16, Santa Cruz #SC-16; 1:1000), Erk2 (C-14, Santa Cruz #SC-154; 1:1000), pERK1/2 (Thr202/204) (Cell Signaling #9101; 1:1000), GAPDH (Abcam #ab8245; 1:5000), GFP (Santa Cruz #SC-9996; 1:1000), Gq/11 (C19, Santa Cruz #SC-392; 1:1000), Gq (E-17, Santa Cruz #SC-393; 1:1000), GRK2 (c-15, Santa Cruz #SC-562; 1:1000), HA (F-7, Santa Cruz #SC-7392; 1:1000), LAMP1 (1D4B, Hybridoma Bank; 1:3000), LC3B (Cell Signaling #2775; 1:1000), LC3B (NB100-2220, Novus Biologicals) mTOR (7C10, Cell signaling #2983; 1:1000), p62 (Progen, GP62; 1:1000), p62 (BML-PW9860, Enzo; 1:1000) p70-S6K (Cell Signaling #2708; 1:1000), p-p70-S6K (Thr389) (Cell Signaling #9205; 1:1000), Raptor (24C12) (Cell signaling, #2280; 1:1000), S6 ribosomal protein (Cell Signaling #2217; 1:2000), p-S6 ribosomal protein (Ser240/244) (Cell Signaling #2215; 1:1000), Ubiquitin (Sigma-Aldrich #U5379; 1:50), ULK1 (D8H5, Cell Signaling #8054; 1:1000), p-ULK1 (Ser-757) (Cell Signaling #6888; 1:1000). Secondary antibodies coupled to HRP (Nordic Immunology): GAR/IgG(h + L)/PO (16461, 1:50,000) and GAM/IgG (H + L)/PO (6513, 1:50,000) and secondary antibodies conjugated to infra-red dyes (LI-COR): Guinea pig 800 nm (926-32411, 1:15,000), Rabbit 800 nm (926-32211, 1:15,000) or 680 nm (926-68021, 1:15,000) and Mouse 800 nm (926-32212, 1:15,000) or 680 nm (926-68022, 1:15,000).

*Immunofluorescence.* Cells previously seeded in glass coverslips, pre-coated with polylysine 1× (Sigma), were fixed in 4% paraformaldehyde (15 min). Aldehyde quenching was carried out by adding 10 mM glycine in PBS for 5 min and permeabilization with Triton X-100 at 0.5% in PBS twice (10 min), following by blocking with 1% BSA for 1 h at RT. Cells were mounted with Fluoromount-G-Dapi (Southern Biotech). Sample images were acquired using a confocal laser microscope LSM710 (Zeiss) at ×63 magnification. Image J software (NIH) was used for general image analysis and manipulations. Imaging of the mCherry-GPF-LC3 reporter was carried out using high-content microscopy. Cells were plated in a glass-bottom 96-well plate and after incubation for the desired time, fixed with 4% paraformaldehyde, and images were captured with a high-content microscope (Operetta system, Perkin Elmer) Quantification was performed with the manufacturer's software in a minimum of 2000 cells (~9 fields). The total number of autophagic vacuoles (AV, autophagosomes plus autolysosomes) was estimated from the number of mCherry-positive puncta, whereas puncta positive for GFP and mCherry were scored as autophagosomes (APG). Autophagic flux was calculated as the number of mCherry only fluorescent puncta (autolysosomes, AL) per cell.

*Quantification of lysosome distribution.* Multi-channel fluorescence images for DNA (Dapi), rat anti-Lamp1 (using Alexa 555, goat anti-rat IgG, ThermoFisher, as a secondary antibody), and a cell tracer (Alexa 488, goat anti-mouse IgG1, ThermoFisher) were acquired in a Zeiss LSM780 confocal microscope equipped with a ×63 Oil (NA = 1.4) objective (MIP-IBMB). Cells were blindly selected using the

green channel and, for each, a Z-stack was performed using $50 \times 50 \times 500$ nm voxel size.

Before quantification, the maximum intensity projection was obtained for each cell image, and the cell and nucleus boundaries were extracted by respectively manual and automatic segmentation. On the red channel, the background was subtracted using the rolling ball algorithm (radius = 30 pixels), and the lysosome distribution was then analyzed using an original ImageJ/Fiji script developed to that purpose, available at https://github.com/MolecularImagingPlatformIBMB/ringIntensityDistribution.[63] Taking as reference the cell boundary, each cell was divided in four concentric rings of equivalent area, being each ring isometrically centered at the nucleus centroid. The fluorescence intensity density per ring was then measured and normalized with respect to the total intensity density of the cell (see Appendix Fig. S4 for details). At least 30 cells were measured for each condition.

*xCELLigence measurements.* The xCELLigence system RTCA SP instrument (Roche Applied Science) monitors changes in the cell index (a measure of cell attachment to the plate) which has been shown to effectively correlate to proliferation, adhesion, and viability changes[64]. To assess proliferation cells were seeded in a 96-well gold electrode sensor plate (E-plates 16) and monitored every 15 min for varying times (50–100 h) in 10% FBS. Cell index (CI) values were normalized 7 h after plating (to exclude the adhesion phase). Cell growth was calculated as the slope (hours-1) of the cell index curve during a total of 50 h. In no case was cell death due to excessive confluence as confirmed by plate inspection with a microscope.

*Annexin V/7-AAD binding.* To quantitatively measure apoptosis, the PE Annexin V Apoptosis Detection kit I (BD-Bioscience) was utilized. Gαq/11 WT and Gαq/11 KO MEFs at 90% confluency were starved during 24 h with 0.1% FBS after which cells were re-suspended in Annexin V-binding buffer (0.1 M HEPES/NaOH (pH 7.4), 1.4 M NaCl, 25 mM $CaCl_2$). Samples were analyzed by flow cytometry on a BD FacsCalibur flow cytometer (BD-Bioscience). To determine the apoptotic stage of the different cell populations, 7-AAD- and Annexin V-positive cells were determined with the CellQuest Pro Version 4.0 Software (BD-Bioscience) and analyzed with the FlowJo Software (v7.6.5 and v10.0.8r1). Cells treated with staurosporine (2.5 μM, 2 h) or ultraviolet irradiation (2 h) were considered the apoptotic (annexin V-positive) and necrotic (7-AAD-positive) controls, respectively.

*Autophagy detection in living cells by flow cytometry.* The autophagic activity of serum, amino acids, or glucose-starved WT and Gαq/11 KO MEF cells was monitored using Cyto-ID Autophagy Detection Kit (Enzo Life Sciences) according to the manufacturer's instructions. Treatment with chloroquine (50 μM, Sigma-Aldrich) during the last 4 h of the serum starvation treatment or during 4 h before the amino acids or glucose removal, allowed quantifying the autophagic flux. Samples were analyzed by flow cytometry on a BD FacsCalibur flow cytometer (BD-Bioscience), and data were analyzed with the FlowJo software.

*Transmission electron microscopy and morphometric analysis.* For conventional TEM, Gαq/11 WT and Gαq/11 KO MEFs grown at 10% FBS were fixed in situ for 2 h at room temperature with a mixture of 4% PFA and 2% glutaraldehyde in 0.1 M Sörensen phosphate buffer (pH 7.4). After several washes, cells were scraped and collected in the same buffer, centrifuged and the cell pellets were processed for embedding in the Epoxy, TAAB-812 resin (TAAB Laboratories) according to standard procedures[65]. Ultrathin sections (70–80 nm) of the cell samples were stained with 2% uranyl acetate solution in water and lead Reynolds citrate and examined at 80 kV in a Jeol JEM-1010 (Tokyo, Japan) electron microscope. Pictures were taken with a TemCam-F416 (4 K × 4 K) digital camera (TVIPS). Autophagic vacuoles were identified using established criteria[59]. Autophagic vacuoles (vesicles of 0.5 μm) were classified as autophagosomes when they met two or more of the following criteria: double membranes (complete or at least partially visible), absence of ribosomes attached to the cytosolic side of the membrane, luminal density similar to the cytosol, and identifiable organelles or regions of organelles in their lumen. Vesicles of similar size but with a single membrane (or <40% of the membrane visible as double), luminal density lower than the surrounding cytosol, or multiple single membrane-limited vesicles containing light or dense amorphous material were classified as autophagolysosomes. Total autophagy vacuoles were composed of the sum of autophagosomes and autolysosomes. The maturation of autophagy vacuoles was calculated as the percent of autolysosomes and autophagosomes out of the total number of autophagy vacuoles.

*Isolation of subcellular fractions.* Autophagic vacuoles were isolated from rat liver using discontinuous metrizamide density gradient[66]. Briefly, after homogenization, livers were subjected to sequential differential centrifugations to separate a fraction enriched in autophagy-related compartments and mitochondrial. This fraction was placed at the bottom of a discontinuous (three layers) gradient of metrizamide and upon centrifugation, mitochondria remained at the bottom whereas a fraction enriched in autophagosomes is recovered in the top band and in autolysosomes in the second band from the top. A lysosomal enriched fraction is also recovered from

the third band of the gradient from the top. Upon extensive washing with 0.25 M sucrose, fractions are systematically analyzed for protein markers and lysosomal enzymatic activity[67]. Cytosolic fractions were obtained by centrifugation for 1 h at 100,000×g of the supernatant obtained after separating the mitochondria–lysosome– enriched fraction.

## Quantification and statistical analysis

*Quantification.* Immunoreactivity bands were quantified by laser densitometry with a Biorad GS-900 scanner and using de Bio-Rad provided Image Lab 5.2 (https://www.bio-rad.com) or by the software Application Software Version 3.0 LI-COR (Biosciences), included in the Odyssey Infra-red Imaging System.

Flow cytometry data were analyzed with the FlowJo Software V7.6.5 and V10.0.8r1 (https://www.flowjo.com).

Image J software v1.53j (NIH) was used for general image analysis and manipulations (https://imagej.nih.gov/ij/index.html).

Excel (v1905) was used for data analysis quantification.

*Statistics.* All data are presented as mean values ± SEM of the indicated number of independent experiments stated in the figure legend. We determined the statistical significance in instances of single comparisons by two-sided unpaired Student's *t*-test and in instances of multiple means comparisons by one-way ANOVA followed by the Bonferroni's test as specified in each figure legend. All statistics were obtained using GraphPad Prism 8 software (https://www.graphpad.com/). The differences were considered significant when *$P < 0.05$, **$P < 0.01$, ***$P < 0.001$, and ****$P < 0.0001$.

**Reporting summary.** Further information on research design is available in the Nature Research Reporting Summary linked to this article.

## Data availability

List of Figures that have associated raw data are Figs. 1–7 and Supplementary Figs. 1–11. Other data that support the findings of this study are available from the corresponding authors upon reasonable request. No datasets were generated or analyzed during this study. All unique materials generated are readily available from the authors. Source data are provided with this paper.

## Code availability

An original Fiji macro has been developed and deposited in a Github repository (https://doi.org/10.5281/zenodo.4815136).

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

## Acknowledgements

We thank Paula Ramos, Susana Rojo-Berciano, and Laura López for helpful technical assistance. Dr. Marta Cruces (Universidad Autónoma de Madrid, Spain) for her invaluable help regarding the liver explants experiments, Dr. Badford Berk (University of Rochester, NY, USA) for providing the GFP-Flag-PB1-p62 plasmid, Drs. Stefan Offermanns and Nina Wettschureck (Max-Planck-Institute for Heart and Lung Research, Germany) for providing Tie2-CreERT2; Gnaq f/f; Gna11−/− [EC-q/11-KO] mice, and Dr. Guzmán Sánchez for scientific advice. We thank also Ricardo Ramos from the Genomic facility of Fundación Parque Científico de Madrid (Universidad Autónoma de Madrid, Spain) and Gemma Rodríguez-Tarduchy from the Genomic facility of the Instituto de Investigaciones Biomédicas "Alberto Sols" for their help with cell lines authentication. The help from CBMSO Animal Care, Flow Cytometry, Electron and Optical and Confocal Microscopy facilities is also acknowledged. This work was supported by Ministerio de Economía; Industria y Competitividad (MINECO) of Spain (grant SAF2017-84125-R to F.M.), (grant BFU2017-83379-R to A.M.A.), Instituto de Salud Carlos III (PI18/01662 to CR, co-funded with European FEDER contribution), CIBERCV-Instituto de Salud Carlos III, Spain (grant CB16/11/00278 to F.M., co-funded with European FEDER contribution), Fundación Ramón Areces (to C.R. and F.M.) and Programa de Actividades en Biomedicina de la Comunidad de Madrid-B2017/BMD-3671-INFLAMUNE to F.M. and NIH grants AG021904 and AG038072 to A.M.C. We also acknowledge the support of a Contrato para la Formación del Profesorado Universitario (FPU13/04341) and (FPU14/06670), an EMBO short-term fellowship (ASTF 600-2016). We also acknowledge institutional support to the CBMSO from Fundación Ramón Areces.

## Author contributions

S.C. and M.S.F. designed and performed experiments, prepared figures, analyzed the data, wrote and revised the manuscript; A.C., I.T., A.D., and A.M.A. performed experiments, analyzed the data, and prepared figures. E.R. and A.M.A. designed and E.R. programmed the lysosomal distribution quantification imaging assay, A.M.C. helped with study concepts and revised the manuscript, F.M. Jr and C.R. provided study concept and design, interpreted experiments, supervised the study, wrote and revised the manuscript, and obtained funding.

## Competing interests

A.M.C. consults for Neuropore (La Joya, CA, USA) and is co-founder of Selphagy Therapeutics (Boston, MA, USA). The remaining authors declare no competing interests.
