## [Peer Review File · Nature Communications]

REVIEWER COMMENTS

Reviewer #1 (Remarks to the Author):

Cabezudo and coworkers present a manuscript entitled „Gq controls autophagy via modulation of the mTORC1 signaling hub“. Key to this modulation is the interaction between Gq and p62 via a non-canonical interaction surface. The manuscript contains tons of biological data, is well executed and well presented and except for usage of many abbreviations good to read and logical to follow. My major concern with this manuscript is the novelty claims made by the authors as detailed below.

Lines 124-128: The authors state...„In our search for new effectors and cellular functions of the ubiquitous Gq/11 subunits of heterotrimeric G proteins, which are coupled to many GPCR 18, we unveiled their unanticipated role as a general modulator of autophagy in response to different types of nutrients, by playing a role in the assembly of active mTORC1 multi-molecular complexes.“ This sentence implies that Gq involvement in autophagy is unanticipated and entirely novel. However, Gq involvement in autophagy has been reported in the literature on several occasions. Please see for example „Activation of Gq in Cardiomyocytes Increases Vps34 Activity and Stimulates Autophagy. Liu S, Jiang YP, Ballou LM, Zong WX, Lin RZ J Cardiovasc Pharmacol. 2017 Apr; 69(4):198-211.“ Here, a link between Gq and autophagy has already been established. Moreover, in a Cell press review (Trends in Endocrinology and Metabolism, May 2014, Vol.25, No.5, Wauson et al, there is quite some info on GPCR regulation of autophagy including Gq-coupled receptors). Even links between muscarinic M3, endothelin receptors and mTORC1 are give there already. For these reasons I am not sure whether the novelty claims really hold. Certainly the findings are very interesting and the study is well executed and sound, but is Gq regulation of autophagy really unanticipated? At least this reviewer is aware of several links between Gq and autophagy that exists since several years.

Lines 182-185: „Overall, the unexpected location of Gq in autophagic compartments and lysosomes and the fact that this protein is not degraded under nutrient stress conditions in these organelles suggested a functional role for Gq as an autophagy modulator.“ This reviewer appreciates the unexpected location of Gq in autophagic compartments and the nice EM images, but is, again, puzzled about the suggestion of Gq being a functional modulator of autophagy. This link already exists in the literature. Again, my problem is the claim of apparent novelty not the well done experiments.

Maybe it would be more appropriate to report on the role of Gq/11 as modulator of autophagy kinetics in different situations of nutrient stress? Would this better reflect the novelty of this study?

Discussion 446-448: „Our data identify Gq/11 as a novel modulator of the mTORC1 signaling hub in the control of autophagy.“ Again, Gq and mTORC have been linked in prior literature already.

Minor points:

Abstract: to make the article more accessible for a broad audience avoid the abbreviation p62/SqSTM1

Introduction, line 83: membranes of varied origins form phagophores....please specifically state the nature of the membrane origin and cite relevant literature in support of this.

Line 85: what is LC3B, LC3-II? Explain briefly to enhance accessibility of the article

In general fewer abbreviations will help to make the interesting article easier to access but this may not be necessary for a more specialized journal.

Reviewer #2 (Remarks to the Author):

Cabezudo and others identify Gq/11 as a regulator of autophagy and mTORC1. Several lines of evidence have been presented to support their conclusion. Using MEFs, they found that deficiency in

Gaq/11 promotes autophagy accompanied by decreased mTORC1 signaling. Activation of Gaq/11 also leads to mTORC1 activation and reverts the autophagic phenotype. During nutrient recovery, Gaq/11 was required for mTORC1 reactivation and repress autophagy. They also found that Gaq is part of the mTOR/p62/raptor complex and that Gaq associates with p62. Finally, they showed that mTORC1 reactivation and autophagy modulation by Gaq correlates with the Gaq/p62 interaction.

This is an interesting study that provides new insights on how Gaq controls mTORC1 signaling and autophagy. There are a few reports on how GPCR could be involved in mTORC1 signaling but the current study has carefully analyzed the role of Gaq/11 in promoting mTORC1 signals while suppressing autophagy and also revealed a nutrient-sensitive interaction of Gaq/11 and mTOR with p62. I only have a few comments and questions below.

1. Fig. 3, The authors conclude that activation of Gaq reverts the autophagic phenotype promoted by nutrient stress conditions. In Figure 3 B, while there is an increase in p62, the changes in LC3I/II levels do not seem to support their conclusion. Furthermore, in Fig 3C. the addition of CNO also decreased LC3II levels at 16-24 hr (p62 levels were not shown). The effect of CNO addition upon amino acid starvation on autophagy repression was also not shown in Fig 3D. Hence, more convincing data to support such autophagy reversion upon Gaq stimulation is needed to support their conclusion in this figure.
2. The addition of CNO in Fig 3 c and 3d diminished S6 phosphorylation at 0 time point. Can the authors comment on this? Is the Gaq/11 stimulation of mTORC1 signaling dependent on growth factor/PI3K signaling?
3. Does the Gaq/11-mediated increase in pS6/pS6K phosphorylation blocked by treatment with rapamycin or raptor knockdown?
4. Is mTORC1 signaling/autophagy repression also modulated by other Gaq/11 agonists?

Other comments:

1. The discussion is well written. In the results section, the authors may want to move Fig 6f as part of Fig 7. The discussion of this figure is actually included on the section with other Fig 7 panels.

Reviewer #3 (Remarks to the Author):

This is a very interesting paper reporting the potential role of Gq signaling in mediating nutrient response of mTORC1 signaling. The authors convincingly showed that Gq suppresses basal autophagy, and that Gq has a role in controlling mTORC1 signaling in response to nutrients. Experimental evidence supporting this link seems to be strong. Less obvious part is the molecular mechanism underlying this regulation, which needs to be substantiated by additional experiments.

1. mTORC1 is regulated by many independent pathways. It is well characterized that growth factors and nutrients regulate mTORC1 through Rheb and Rag GTPases. Gq signaling is previously implicated in PI3K-AKT pathways. Supplementary Fig.7a indeed indicates that Gq activation first activates AKT and then mTORC1/S6K. This actually suggests that Gq activates mTORC1 through AKT, in addition to the author's proposed mechanism. Should the authors treat the cells with PI3K/AKT inhibitor and see if Gq control of mTORC1 is maintained?

2. The role of p62 in mediating the Gq effect is interesting; however, is not convincingly proven. Additional experiments are necessary to substantiate this.

2-1) The authors need to inspect whether GaqQL-AA, Gaq-Y261F, Gaq-T257E and Gaq-W263D are still able to control mTORC1 and autophagy. This was only partially done in Supplementary Fig 10, and the

results indicate that their activities are different from Gqwt

2-2) Supplementary Fig 10A -- U73XXX inhibitors did not seem to work. ERK seems to be still robustly activated. The authors need to examine AKT signaling, which has a more direct role in mTORC1 regulation. p-S6 seems to be rather inhibited, which is strange. This is an important figure and needs to be presented in the main figure.

2-3) p62 should be genetically modulated (e.g. knockout or PB1 mutant knock-in) to assess whether the p62 angle is indeed important for Gq action.

3. For most figures, only p-S6 was used as a proxy of mTORC1 activity. More markers are required, such as p-S6K and p-4EBP.

POINT BY POINT RESPONSES TO REVIEWERS

REVIEWER 1

Cabezudo and coworkers present a manuscript entitled „Gaq controls autophagy via modulation of the mTORC1 signaling hub“. Key to this modulation is the interaction between Gaq and p62 via a non-canonical interaction surface. The manuscript contains tons of biological data, is well executed and well presented and except for usage of many abbreviations good to read and logical to follow. My major concern with this manuscript is the novelty claims made by the authors as detailed below

We appreciate these overall positive comments of the reviewer.

MAJOR POINTS

Lines 124-128: The authors state...„In our search for new effectors and cellular functions of the ubiquitous Gaq/11 subunits of heterotrimeric G proteins, which are coupled to many GPCR 18, we unveiled their unanticipated role as a general modulator of autophagy in response to different types of nutrients, by playing a role in the assembly of active mTORC1 multi-molecular complexes.“ This sentence implies that Gq involvement in autophagy is unanticipated and entirely novel. However, Gq involvement in autophagy has been reported in the literature on several occasions. Please see for example „Activation of Gaq in Cardiomyocytes Increases Vps34 Activity and Stimulates Autophagy. Liu S, Jiang YP, Ballou LM, Zong WX, Lin RZ J Cardiovasc Pharmacol. 2017 Apr; 69(4):198-211.“ Here, a link between Gq and autophagy has already been established. Moreover, in a Cell press review (Trends in Endocrinology and Metabolism, May 2014, Vol.25, No.5, Wauson et al, there is quite some info on GPCR regulation of autophagy including Gq-coupled receptors). Even links between muscarinic M3, endothelin receptors and mTORC1 are give there already. For these reasons I am not sure whether the novelty claims really hold. Certainly the findings are very interesting and the study is well executed and sound, but is Gq regulation of autophagy really unanticipated? At least this reviewer is aware of several links between Gq and autophagy that exists since several years.

Lines 182-185: „Overall, the unexpected location of Gaq in autophagic compartments and lysosomes and the fact that this protein is not degraded under nutrient stress conditions in these organelles suggested a functional role for Gaq as an autophagy modulator.“ This reviewer appreciates the unexpected location of Gq in autophagic compartments and the nice EM images, but is, again, puzzled about the suggestion of Gq being a functional modulator of autophagy. This link already exists in the literature. Again, my problem is the claim of apparent novelty not the well done experiments.

Maybe it would be more appropriate to report on the role of Gaq/11 as modulator of autophagy kinetics in different situations of nutrient stress? Would this better reflect the novelty of this study?

Discussion 446-448: „Our data identify Gaq/11 as a novel modulator of the mTORC1 signaling hub in the control of autophagy.“ Again, Gq and mTORC have been linked in prior literature already.

We are aware of previous reports in the literature suggesting a link between GPCR and Gq-triggered pathways and the modulation of autophagy. In fact, the review by Wauson et al. (Trends End. Metab., 2014) mentioned by the reviewer was cited in the introduction (ref. number 17), and related original work by the same group was cited in the Discussion section (Wauson et al., Autophagy, 2013, former ref. number 39; Wauson et al., Mol Cell., 2012, former ref. number

45). However, as also pointed by the reviewer, our manuscript provides new insights on the role of Gq in the modulation of autophagy in different situations of nutrient stress and identifies new molecular mechanisms linking Gq to this process, acting as a core component of mTORC1 complex via a nutrient-sensitive interaction with p62 by a non-canonical interaction surface.

In sum, we agree with the reviewer in revising the sentences indicated in his/her report in order to better reflect the novelty aspects of our study. We have modified to this end wording of the abstract, the last paragraph of the introduction (incorporating also here the reference by Liu et al., 2017 mentioned by the reviewer), the last sentence in the “Gaq localizes in lysosomal and autophagic compartments” Results section, and the first sentences of the second paragraph of the Discussion.

Minor points:

Abstract: to make the article more accessible for a broad audience avoid the abbreviation p62/SqSTM1

The abbreviation SQSTM1 has been removed from the Abstract as suggested by the reviewer

Introduction, line 83: membranes of varied origins form phagophores...please specifically state the nature of the membrane origin and cite relevant literature in support of this.

As suggested, the varied origins of the membranes forming the phagophores are now specifically mentioned and relevant literature citations more clearly placed in the text.

*Line 85: what is LC3B, LC3-II? Explain briefly to enhance accessibility of the article
In general fewer abbreviations will help to make the interesting article easier to access but this may not be necessary for a more specialized journal.*

We used LC3B to refer to the full-length protein in the immunofluorescence analysis since the antibodies do not differentiate the cleaved and lipidated form (LC3-II) from the unmodified protein. However, we agree with the reviewer on the need to clarify and simplify to avoid reader's confusion. We have now stated more clearly at first use what LC3-II refers to and only refer to LC3-II in the immunoblots for clarity.

REVIEWER 2

Cabezudo and others identify Gaq/11 as a regulator of autophagy and mTORC1. Several lines of evidence have been presented to support their conclusion. Using MEFs, they found that deficiency in Gaq/11 promotes autophagy accompanied by decreased mTORC1 signaling. Activation of Gaq/11 also leads to mTORC1 activation and reverts the autophagic phenotype. During nutrient recovery, Gaq/11 was required for mTORC1 reactivation and repress autophagy. They also found that Gaq is part of the mTOR/p62/raptor complex and that Gaq associates with p62. Finally, they showed that mTORC1 reactivation and autophagy modulation by Gaq correlates with the Gaq/p62 interaction.

This is an interesting study that provides new insights on how Gaq controls mTORC1 signaling and autophagy. There are a few reports on how GPCR could be involved in mTORC1 signaling but the current study has carefully analyzed the role of Gaq/11 in promoting

mTORC1 signals while suppressing autophagy and also revealed a nutrient-sensitive interaction of Gq/11 and mTOR with p62. I only have a few comments and questions below.

We appreciate these overall positive comments of the reviewer.

1. Fig. 3, The authors conclude that activation of Gq reverts the autophagic phenotype promoted by nutrient stress conditions. In Figure 3 B, while there is an increase in p62, the changes in LC3I/II levels do not seem to support their conclusion. Furthermore, in Fig 3C. the addition of CNO also decreased LC3II levels at 16-24 hr (p62 levels were not shown). The effect of CNO addition upon amino acid starvation on autophagy repression was also not shown in Fig 3D. Hence, more convincing data to support such autophagy reversion upon Gq stimulation is needed to support their conclusion in this figure.

Figure 3b. In these experiments, cells growing in 10% serum were starved for 16h (so downmodulation of mTORC1 pathway and upregulation of autophagy takes place), followed by CNO-mediated stimulation of Gq pathways. Although widely used as readouts, the interpretation of dynamic changes in LC3I/II levels is not always straightforward and may vary with the particular experimental conditions, since both conversion of LC3-I to LC3-II and degradation of this protein can take place (Mizushima & Yoshimori, *Autophagy*, 3:6, 542-545, 2007). To address the comment of the reviewer, we have performed additional experiments in this setting. A **new Fig.3b panel** is now shown that allows comparison with the 10% serum initial conditions and more clearly indicates that prolonged stimulation with CNO fosters a sustained activation of the mTORC1 cascade (as assessed by the p-S6 readout) eventually leading to inhibition of autophagy (decreased LC3-II and increased p62 levels) to reach values comparable to the 10% serum condition.

Fig. 3c. We apologize if we failed to provide a clear explanation and that lead to some level of confusion. The LC3I/II data shown in Fig. 3c are in fact consistent with the proposed effects of CNO, whereby a time-dependent increase in mTOR signaling (increased levels of p-S6) associates with gradual decrease in LC3-II levels and this inhibitory effect on autophagy follows faster dynamics than in absence of CNO. We now provide the following more detailed description of the data in the Results section:

After 24h in low serum conditions (time 0 in the immunoblots shown in Fig. 3c), cells displayed the characteristic features of autophagy induction (inactivation of the mTORC1 pathway and increment of LC3-II levels compared to the 10% FBS control situation) (Fig. 3c). In control cells (without CNO stimulation), this pattern persists in the subsequent hours, with decreased levels of p-S6 (Fig.3c) and pS757-ULK1 (Suppl.Fig.7b) (as mTORC1 cascade readouts) and high LC3-II levels (Fig.3c), and only slowly reverts at 16-24h, likely via the nutrients released by the autophagic process. Instead, the specific activation of Gq/11 by CNO led to more marked and earlier (1-4h) increase in p-S6 and pS757-ULK1 (Fig. 3c, and Suppl. Fig.7b), also leading to an earlier and stronger inhibition of autophagy, as detected by more rapidly decreasing levels of LC3-II at 1-4h of CNO treatment compared to control non-Gq-stimulated cells (Fig. 3c).

In sum, CNO addition leads to an earlier decrease of LC3-II levels compared to control, as it would be expected for autophagy repression via enhanced

mTORC1 signaling (pS6 readout). To further support this point, we have performed additional experiments that show a similar pattern of autophagy modulation when testing the evolution of p62 levels (**new Suppl. Fig. 7c**).

Fig.3d (now re-numbered 3e in the new version). As suggested by the reviewer, we have performed additional experiments in these conditions that show that CNO-mediated preservation of mTORC1 activation also attenuated the increase in LC3-II levels triggered by amino acid retrieval in control cells (**new lower panel** in Fig.3e and **new Suppl. Fig.7h**).

Overall, we think that these new data, along with the evidence obtained in other experimental systems such as MEFs shown in other sections of the manuscript, support the proposed role of Gαq in autophagy modulation. We thank the reviewer for his/her suggestions in order to improve this figure.

2. The addition of CNO in Fig 3 c and 3d diminished S6 phosphorylation at 0 time point. Can the authors comment on this?

As detailed above in our response to Fig.3c, the 0 time point represent cells starved for 24h in 0.1% FBS (Fig.3c) or in amino-acid-free EBSS medium (former Fig.3d, now 3e), so pS6 phosphorylation is expected to be diminished compared to the 10% serum conditions because of the absence of nutrients. Addition of CNO at the 0 time point leads to enhanced pS6 phosphorylation at later time points (from 1 h on, Fig.3c) or to prevent the further decrease in pS6 observed in control cells kept in EBSS medium at 30 min (Fig.3e). A similar pattern is observed in these settings for other readouts of mTORC1 pathway activation as p-ULK1 or pp70S6K (Suppl.Fig.7b and f). We have tried to better explain these results in the text.

Is the Gαq/11 stimulation of mTORC1 signaling dependent on growth factor/PI3K signaling?

Data in Fig.3c and Suppl.Fig.7b show that CNO can trigger robust mTORC1 pathway stimulation in the presence of 0.1% serum. The expression of Gαq constructs can also preserve the activation of this cascade in low serum conditions (Suppl. Fig. 7i and Fig. 6a-b in the new figure numeration).

Regarding the implication of the PI3K/Akt pathway, in the serum context shown in Fig.3c, after the decline in Akt activation promoted by starvation, we observed a slight increase in Akt stimulation at longer times (4 to 24h) in both conditions, while in contrast, we did not observe recovery of Akt activation 30 min after EBSS (Fig. 3e). The fact that the reactivation of the mTORC1 pathway triggered by CNO is observed at much earlier times and does not parallel the pattern of Akt activation, led us to suggest that different upstream mTORC1 modulators would be involved in the effects of Gαq/11.

However, in order to more clearly address this important point, we have performed additional experiments (**new Fig.3d and Suppl. Fig.7d**) that show a clear CNO-dependent modulation of the mTORC1/autophagy cascade even in the presence of specific Akt inhibitors, both in 10% or 0.1% serum conditions.

We thank the reviewer for his/her suggestions regarding this point that have contributed to further strengthen our manuscript.

3. Does the Gαq/11-mediated increase in pS6/pS6K phosphorylation blocked by treatment with rapamycin or raptor knockdown?

Following the suggestion of the reviewer, we now show that CNO/Gαq - dependent modulation of the mTORC1 cascade (pS6 readout) is markedly attenuated by known mTOR kinase inhibitors such as rapamycin (**new Suppl. Fig.7e**)

4. Is mTORC1 signaling/autophagy repression also modulated by other Gαq/11 agonists?

Our data in different figures and using different approaches (KO MEFs, Gq mutants, YM-mediated inhibition) indicate that Gαq/11 is required for the modulation of mTORC1 and autophagy in response to a variety of stimuli (serum, amino acids, glucose), that would act as upstream Gαq/11 agonists in these contexts as pointed out in the Discussion section. The data obtained using the CNO system also suggest that plasma membrane GPCR coupled to Gαq/11 can lead to the modulation of these pathways. However, whether this is a general feature of all GPCR-Gαq/11 agonists in physiological settings is an interesting issue that awaits further investigation. As detailed in the Discussion section, certain ubiquitous Gq-coupled GPCRs reported to sense amino acids and/or glucose, including the taste receptors T1R1/T1R3, the Calcium Sensing Receptor (CaSR), GPRC6A or metabotropic glutamate receptors, or receptors for serum components as lysophosphatidic acid or sphingosine-1-phosphate emerge as particularly interesting candidates for future research.

Other comments:

1. The discussion is well written. In the results section, the authors may want to move Fig 6f as part of Fig 7. The discussion of this figure is actually included on the section with other Fig 7 panels.

We thank the comments of the reviewer. Following her/his suggestion, and also to re-structure the presentation of some results to accommodate comments by other reviewer, we have moved former Fig.6f-g panels to Fig.7 a-b, what we agree allows for a more integrated discussion with data in other Fig.7 panels.

REVIEWER 3

This is a very interesting paper reporting the potential role of Gq signaling in mediating nutrient response of mTORC1 signaling. The authors convincingly showed that Gq suppresses basal autophagy, and that Gq has a role in controlling mTORC1 signaling in response to nutrients. Experimental evidence supporting this link seems to be strong. Less obvious part is the molecular mechanism underlying this regulation, which needs to be substantiated by additional experiments.

We appreciate these overall positive comments of the reviewer.

1. mTORC1 is regulated by many independent pathways. It is well characterized that growth factors and nutrients regulate mTORC1 through Rheb and Rag GTPases. Gq signaling is previously implicated in PI3K-AKT pathways. Supplementary Fig.7a indeed indicates that Gq activation first activates AKT and then mTORC1/S6K. This actually suggests that Gq activates mTORC1 through AKT, in addition to the author's proposed mechanism. Should the authors treat the cells with PI3K/AKT inhibitor and see if Gq control of mTORC1 is maintained?

Regarding the implication of the PI3K/Akt pathway, we agree with the reviewer in that one possible interpretation of the data in Suppl. Fig.7a is that prior activation by Gq of AKT would be involved in the stimulation of the mTORC1/S6K pathway at later time points. In the serum-removal context shown in Fig.3c, after the decline in Akt activation promoted by starvation, we observed a slight increase in Akt stimulation at longer times (4 to 24h) in both conditions, while we did not observe Akt activation recovery 30 min after EBSS (renumbered Fig. 3e). The fact that the reactivation of the mTORC1 pathway triggered by CNO is observed at much earlier times and does not parallel the pattern of Akt activation, led us to suggest that different upstream mTORC1 modulators would be involved in the effects of Gq/11.

However, in order to more clearly address this important point, as suggested by the reviewer we have performed additional experiments (**new Fig.3d and Suppl.Fig.7d**) that show a clear CNO-dependent modulation of the mTORC1/autophagy cascade even in the presence of specific Akt inhibitors, both in 10% or 0.1% serum conditions. We also now show that CNO/Gq - dependent modulation of the mTORC1 cascade (pS6 readout) is markedly attenuated by known mTOR kinase inhibitors such rapamycin (**new Suppl. Fig.7e**).

We thank the reviewer for his/her suggestions regarding this issue.

2. The role of p62 in mediating the Gq effect is interesting; however, is not convincingly proven. Additional experiments are necessary to substantiate this.

2-1) The authors need to inspect whether GqQL-AA, Gq-Y261F, Gq-T257E and Gq-W263D are still able to control mTORC1 and autophagy. This was only partially done in Supplementary Fig 10, and the results indicate that their activities are different from Gqwt

As suggested by the reviewer, we have performed new experiments to test the effects of these mutants on the activation of the mTORC1 pathway in low serum conditions, and re-structured the presentation of the data to allow for a more **integrated analysis of the ability of Gq mutants to modulate the mTORC1 cascade compared to their capability to associate with p62.**

New Fig.6a (former Suppl. Fig.10b) shows that expression of the GqQL-AA mutant (that disrupts binding of Gq to its canonical effectors) was able to promote activation of mTORC1 pathway to a significant extent under low serum conditions, whereas overexpression of the Gq-EEAA mutant (unable to interact with PB1-domain-containing Gq effectors) was not able to mimic the effects of Gq WT in such experimental conditions (new Fig. 6b, former Fig.5f). Overall, these results suggested that the PB1-binding region of Gq is involved in mTORC1 pathway modulation.

Consistent with these data and with the notion that p62/ Gαq association mediates mTORC1 modulation via non-canonical cascades, the GαQL-AA mutant was still able to efficiently bind p62 (new Fig.6d, former Suppl.Fig11c), whereas mutations in the PB1-binding region of Gαq markedly decreased association with p62 compared to wild type Gαq (Supplementary Fig.10c), and the GαqRC-EEAA mutant was unable to potentiate the Gαq/p62 complex formation (new Fig. 6e, formerly 6d).

Following the specific suggestion of the reviewer, we then noted that the Gαq mutants (Gαq-Y261F, Gαq-T257E, Gαq-W263D) unable to bind GRK2 co-immunoprecipitated with p62 to a greater extent than wild-type Gαq (new Supplementary Fig.10e, former Suppl. Fig.11d). **Additional experiments (new Supplementary Fig.10f)** show that these mutants were able to sustain mTORC1 pathway activation and modulate autophagy to a higher extent than wild-type Gαq under low serum conditions, again consistent with a close correlation between the ability of interact with p62 and that of fostering mTORC1 stimulation.

2-2) Supplementary Fig 10A -- U73XXX inhibitors did not seem to work. ERK seems to be still robustly activated. The authors need to examine AKT signaling, which has a more direct role in mTORC1 regulation. p-S6 seems to be rather inhibited, which is strange. This is an important figure and needs to be presented in the main figure.

In line with the suggestion of the reviewer, as detailed in our answer to your point 1 above, we have now directly tested the role of the AKt pathway in Gq-mediated mTORC1 activation using specific inhibitors (**new Fig.3d and Suppl.Fig.7d**). Regarding the experiments with the U73 PLCbeta inhibitors, we have decided to remove them and address the role of this pathway in a more specific way by using the GαQL-AA mutant (unable to interact with this and other canonical Gq effectors). In line with the suggestion of the reviewer, now both experiments with Akt inhibitors (Fig.3d) and the GαQL-AA mutant data (Fig.6a, 6d) are presented in main figures.

2-3) p62 should be genetically modulated (e.g. knockout or PB1 mutant knock-in) to assess whether the p62 angle is indeed important for Gq action.

Regarding the potential use of p62 KO or p62-modifying strategies to explore the implication of this protein in Gαq effects on mTORC1/autophagy modulation, although interesting, unfortunately these experiment are not straightforward, since the p62 multidomain protein has been shown to be a key player in this process as well as in several other cellular processes besides autophagy, such as cell growth or oxidative stress response, among others (Sánchez-Martín & Komatsu, J. Cell Sci. 2018; Moscat, et al., Cell, 2016). Therefore, the interpretation of the impact of its silencing or functional modification would be very complex. Since we have characterized that Gαq associates with p62 through its acidic PB1-like domain, and have been able to identify Gαq mutants with decreased (Gα-EEAA) or slightly enhanced (Gαq-Y261F, Gαq-T257E, Gαq-W263D) Gαq/p62 interaction, we believe, in line also with the suggestion

of the reviewer in point 2.1, that the comparative use of these mutants is a suitable, specific, and less disruptive approach to assess the role of Gαq/p62 association in the modulation of autophagy and the mTORC1 pathway.

In this regard, as discussed in our answer to point 2.1, we have added new data with some of these Gq mutants, **and re-structured the presentation of results in new Figs. 6 and 7 and Suppl. Fig. 10 and 11 to more clearly substantiate the role of p62 in mediating Gq effects.** After comparing the ability of Gq mutants to modulate the mTORC1 cascade compared to their capability to associate with p62 (new Fig. 6 and Suppl. Fig.10, see our answer to point 2.1), we present in Fig.7 a and b that the modulation Gαq/p62 association is regulated by nutrient availability. Next, we present the reconstitution experiments in Gαq/11 KO MEFs and MEFs stably expressing in a Gαq/11 KO background either Gαq wt (Gαq/11 KO +Gq MEFs) or the Gαq mutant with decreased Gαq/p62 interaction (+Gq-EEAA). These data indicate that basal activation status of the mTORC1 cascade and autophagy levels correlated with the ability of Gαq mutants to interact with p62 (Fig.7c), consistent with the analysis of endogenous mTOR complexes in these cells showing a decreased association with Gαq and p62 in +Gαq-EEAA MEFs compared to Gαq/11 KO +Gq MEFs (Fig.7d), and with the pattern of data from serum or amino acid starvation/recovery experiments (Fig. 7e and Suppl Fig.11b).

We trust that such re-structured presentation of the results more clearly supports the notion that the interaction between Gαq and p62 in response to nutrients is key for the stimulation of the mTORC1 pathway and the control of autophagy.

3. For most figures, only p-S6 was used as a proxy of mTORC1 activity. More markers are required, such as p-S6K and p-4EBP.

We used the phosphorylation status of the S6 ribosomal protein (a reliable readout of mTORC1 activation used in many reports) in all experiments assessing the status of this pathway. In addition (and within the methodological limitations imposed by the available commercial antibodies for p-p70S6K, total S6K, pS757-ULK1 and total ULK1, which in our hands display variability in their detection ability depending on the batch, the cell type analyzed and the experimental settings), please note that we have already corroborated, when possible, the correlation between p-S6 levels and the activation patterns of its upstream kinase p70-S6K (a direct target of mTORC1) and/or, given the paper's focus on autophagy, the study of p-ULK1 (Ser757) status. Data with these readouts are shown in a variety of Figure panels: Suppl.Fig.3, Suppl. Fig. 6a-b, Suppl.Fig.7a, b,f,g, Suppl. Fig. 8 b,d; Fig.4d)

REVIEWER COMMENTS

Reviewer #1 (Remarks to the Author):

I appreciate the efforts of the authors to further clarify their manuscript and have no more requests.

Reviewer #2 (Remarks to the Author):

The authors have satisfactorily addressed my comments.

Reviewer #3 (Remarks to the Author):

The paper was improved by addressing many of my and other reviewers' comments. However, I still have major issues with the specificity and efficacy of inhibitors. The authors removed U73 PLCbeta inhibitor results, which does not seem to work. They instead included new AKTi work. However, there is still no proof that AKTi actually worked in their system. Some AKT-specific substrates, such as p-FOXO or p-GSK should be used to prove that AKTi was indeed effective. At least based on p-AKT, AKTi does not seem to be effective. The reference for the inhibitor is missing. The authors would need to present more convincing results to highlight that the authors' mechanism is indeed independent of the previously known PI3K-dependent pathways.

NCOMMS-20-15236-A

POINT BY POINT RESPONSES TO REVIEWERS

Reviewer #1 (Remarks to the Author):

I appreciate the efforts of the authors to further clarify their manuscript and have no more requests.

We appreciate the positive comments of the reviewer.

Reviewer #2 (Remarks to the Author):

The authors have satisfactorily addressed my comments.

We appreciate the positive comments of the reviewer.

Reviewer #3 (Remarks to the Author):

The paper was improved by addressing many of my and other reviewers' comments.

We appreciate these overall positive comments of the reviewer.

However, I still have major issues with the specificity and efficacy of inhibitors. The authors removed U73 PLCbeta inhibitor results, which does not seem to work.

We apologize for not having explained this issue in a clearer way to the reviewer in our previous response. The experiments with the U73 PLCbeta inhibitors shown in Suppl. Figure 10A of the first version of the manuscript (see below for your consideration) indicated that after MEF starvation amino acids promoted a robust activation of the mTORC1 pathway as assessed by S6 phosphorylation (circa 3.5-fold). A similar pattern was observed upon addition of the inactive U73343 analog (unable to inhibit PLCbeta), whereas the PLCβ pharmacological inhibitor U73122 had a partial (but not total) inhibitory effect on mTORC1 stimulation (circa 1.75-fold activation of pS6 still observed). Additional quantification of available data (see Figure R2 below for your consideration) confirm this pattern. We therefore concluded that an additional effector was likely involved in Gαq-dependent mTORC1 modulation.

Fig R1 for consideration of the reviewer (previous Suppl Fig. 10A in first version)

Fig. R2 for the consideration of the reviewer. Data from 3 independent experiments performed as in Suppl. Fig.10a of the previous version of the manuscript.

As indicated in our previous response to your comments, even if we believe that the data with the PLCbeta inhibitors support this interpretation, we decided to address the role of this pathway in a more specific way by using the GqQL-AA mutant, unable to interact with PLCbeta and other canonical Gq effectors (Figures 6a, 6d in the revised manuscript). As detailed in the manuscript, this mutant was still able to promote activation of the mTORC1 pathway to a significant extent under low serum conditions, although not as fully as the wild type GqQL construct (Fig.6a), suggesting that an additional effector binding through a different region was involved in Gq-dependent mTORC1 modulation. Consistent with this notion, the GqQL-AA mutant was able to efficiently bind p62 (Fig.6d), the proposed effector linking Gq to mTORC1 pathway modulation.

In sum, we think that the data with the PLCbeta inhibitors and the GqQL-AA mutant converge in pointing to non-canonical effectors being additionally involved in Gq-mediated mTORC1 pathway stimulation. However, given the already high number of panels and figures in our manuscript and the fact that in our view the use of the GqQL-AA mutant is a more specific approach and also addresses the potential implications of canonical effectors other than PLCbeta (as p63RhoGEF), we chose to remove previous Suppl. Fig. 10A from the revised manuscript.

They instead included new AKTi work. However, there is still no proof that AKTi actually worked in their system. Some AKT-specific substrates, such as p-FOXO or p-GSK should be used to prove that AKTi was indeed effective. At least based on p-AKT, AKTi does not seem to be effective. The reference for the inhibitor is missing. The authors would need to present more convincing results to highlight that the authors' mechanism is indeed independent of the previously known PI3K-dependent pathways.

Following the suggestion of the reviewer, we directly addressed in the revised manuscript the role of the Akt pathway in Gq-mediated mTORC1 activation using the specific inhibitor Akti-1/2 (Fig.3d and Suppl.Fig.7d, shown below). Akti-1/2 interacts with the PH domain of Akt and prevents the conformational change required for phosphorylation by upstream kinases such as PDK1 (at T308) and mTORC2 (at S473) and the subsequent Akt activation, thus being widely used and recommended for cellular studies. As suggested by the reviewer, references for this inhibitor have now been included in the manuscript (Bain et al., 2007; Logie et al., 2007).

In control settings (no inhibitor), we find that the CNO/Gαq-cascade markedly stimulates the mTORC1 pathway (pS6 readout), what correlates with a clear inhibition of autophagy as assessed by the LC3II marker), in both 10% (lanes 1-2 of the gel) or 0.1% (lanes 5-6) serum conditions, in the absence of parallel changes in the pT308-Akt phosphorylation status. The same pattern is observed in the presence of the Akti-1/2 inhibitor in both serum conditions (lanes 3-4 and 7-8 of the gel in Fig.3d above).

The reviewer points that these results do not convincingly proof that Akti-1/2 actually worked effectively in our system, at least based on the p-AKT-T308

data shown. We agree that some of the p-T308 data are intriguing. In 10% serum and absence of CNO conditions, a significant reduction in pT308-Akt is noted in the presence of the inhibitor (compare lanes 1 and 3 and quantification of 3 experiments in Figure R3 below), consistent with Akti-1/2 being effective even in the context of the persistent activation of the pathway in high serum situations. However, after 24h in 0.1% FBS conditions, pT308 levels tend to be higher than in 10% FBS and no clear effect of Akti-1/2 is observed (compare lanes 5 and 7 in figure above and quantification of 3 experiments in Figure R3 below).

*Figure R3 for consideration of the reviewer. DREADD-Gq-HEK293 cells growing in 10% FBS or starved for 24h with 0.1% FBS medium were stimulated for 4h with CNO (1µM) or vehicle in the absence or presence of the AKTi 1/2 inhibitor (1µM). Graph only represents the results obtained in absence of CNO conditions. Phospho-AKT (T308) data (mean ± SEM of 3 independent experiments) were normalized using total AKT protein. Statistical significance was analyzed using unpaired t-test, ***p<0.001.*

In search of **alternative bona fide readouts** of Akti-1/2 effectiveness, we have now assessed the **p-S473 Akt phosphorylation status** in the same samples previously analyzed in Fig.3d. Of note, a marked attenuation of the p-S473-Akt activation status was detected in 10% serum and absence of CNO conditions (compare lanes 1 and 3 in Figure R4 below). In 0.1% FBS conditions, pS473 levels were much lower and further decreased with Akti-1/2 (see lanes 5 and 7). The same pattern was observed in two new independent experiments performed in similar settings (see representative gel in Fig. R5 and quantitation in Figure R6).

Figure R4 for consideration of the reviewer: DREADD-Gq-HEK293 cells growing in 10% FBS or starved for 24h with 0.1% FBS medium were stimulated for 4h with CNO (1 μ M) or vehicle in the absence or presence of the AKTi 1/2 inhibitor (1 μ M). The activation of mTORC1 and Akt pathways was analyzed by western blot by assessing the phosphorylation status or S473-AKT or that of downstream targets of mTORC1 (S6 ribosomal protein).

Figure R5 for consideration of the reviewer: Same experimental conditions of Figure R4, new independent experiments

Figure R6 for consideration of the reviewer: Same experimental conditions of Figure R4 and R5. Graph only represents the results obtained in absence of CNO conditions. Phospho-AKT (S473) data (mean \pm SEM of 3 independent experiments) were normalized using total S6 protein. Statistical significance was analyzed using unpaired t-test, ** $p < 0.01$, **** $p < 0.0001$.

In the search for potential explanations of such differential patterns of activation and inhibition of pT308 and pS473 in 10% and 0.1% serum conditions, we noticed that it has been reported that serum starvation increased Akt phosphorylation at T308 in ovarian cancer cells (Dai et al., 2016). Combined serum and glucose deprivation (Gao et al., 2014) also induced selective Thr308 Akt phosphorylation and phosphorylation of a distinct subset of AKT downstream targets in different cells types, including HEK-293 cells, as a possible mechanism to cope with metabolic stress. Of note, it has been reported that Calmodulin kinase kinase 2 (CamKK2) phosphorylates Akt at T308 in a PI3K/PDK1-independent manner (Gocher et al., 2017), whereas other authors suggest the formation of a complex of AKT, PDK1 and the GRP78 chaperone protein in starvation conditions, thus directing phosphorylation of AktThr308 but not AKTSer473 (Gao et al., 2014). Since Akti-1/2 acts by interacting with the PH domain of Akt, this inhibitor might be less effective in preventing T308 phosphorylation by these different mechanisms in such conditions, while still being able to strongly inhibit mTORC2-dependent S473 phosphorylation.

As suggested by the reviewer, we have also investigated GSK3 phosphorylation status as other readout of Akt activity (see Fig. R7 for the consideration of the reviewer). A slight inhibition of p-GSK3 is apparent in low serum conditions and in the presence of Akti-1/2 (comparing lanes 1 and 3 and lanes 5 and 7).

Figure R7 for consideration of the reviewer: Same experimental conditions as in Fig. R4-R6. The activation of mTORC1 and Akt pathways was analyzed by western blot by assessing the phosphorylation status of downstream targets of AKT (p-GSK3 α (S21)/ β (S9)) or of mTORC1 (S6 ribosomal protein). Blot representative of two independent experiments.

However, the pattern is different from the strong inhibition noted using the pS473-Akt readout. It should be noted, however, that the interpretation of the GSK3 phosphorylation levels in relation to Akt activation status is not straightforward. Multiple signaling pathways can also target GSK3 α S21 and GSK3 β S9 phosphorylation, such as PKA, p70 S6 kinase or PKC. Downstream of mTORC1, p70S6K can phosphorylate these GSK-3 sites in the presence of amino acids (Moore et al, 2013; Maurer et al., 2014). Although the overall activity of Akt lacking S473 phosphorylation is greatly diminished (Manning and Tocker, 2017), it has been reported in the context of lack of

nutrients (Gao et al., 2014), that Akt phosphorylated at T308 but not at S473 (as we also observe in 0.1 % FBS conditions) can still phosphorylate GSK3 to a certain extent.

In summary, we believe that our new data using the p-S473 Akt phosphorylation status as a readout are consistent **with a bona fide inhibition of canonical PI3K-dependent Akt stimulation pathways** in the presence of the Akti-1/2 inhibitor, although other alternative pathways leading to partial Akt stimulation via pT308 phosphorylation may be also present in 0.1 % FBS conditions. Although the differential T308 and S473 Akt phosphorylation status observed in some experimental conditions and their potential interactions with the Gq-governed mTORC1-S6 pathways might be of interest and deserves future investigation, we think that incorporating these data might be too complex and is out of the scope of this manuscript.

Overall, our data (new Fig.3d, lower gel) showing marked stimulation of the mTORC1 pathway (pS6 readout) upon activation of the CNO/Gαq -cascade in both 10% or 0.1% serum experimental conditions in the absence of parallel changes in Akt phosphorylation status and also in the presence of Akti-1/2 and lack of p-S473 Akt phosphorylation are consistent with the occurrence of **alternative routes** of Gq-mediated mTORC1 stimulation in addition to previously known canonical PI3K-dependent pathways, a notion reinforced by all the data regarding the p62-Gq-mTORC1 interaction detailed in other figures of the manuscript.

Finally, we want to stress that, as already pointed out in our previous Discussion section (*“the contribution of other mechanisms to Gαq-mediated modulation of mTORC1/autophagy pathways cannot be ruled out.such pathways might also contribute to the overall effect observed, consistent with our findings showing that the modulation of the mTORC1 cascade exerted by Gαq/11 is only partially dependent on its canonical effectors”*), we acknowledge that our new proposed mechanism of Gq-mediated mTORC1 regulation does not rule out the contribution of other Gq-triggered cascades. In line with the comments of the reviewer and given the highly interconnected nature of the mTORC1 and Akt pathways, including the existence of complex feedback mechanisms (Manning and Tocker, 2017), we have rephrased this sentence to more explicitly mention the potential participation of the Akt cascade in Gq mediated mTORC1 stimulation.

In sum, with all these points in consideration, in the revised version of the manuscript, we have rephrased the indicated sentences in the Results and Discussion sections to make these points clearer. In addition, we have incorporated the additional experiments shown in Fig.R5 to **new Fig.3d** (lower panel) to show p-S473 phosphorylation as a more straightforward readout of Akt phosphorylation and effectiveness of the Akti-1/2 inhibitor. Suppl Fig. 7d (shown here below) has also been modified to incorporate the quantification data of these additional experiments and show the significant stimulation by Gq cascades of the mTORC1/S6 pathway in all the experimental conditions.

New Suppl. Figure 7d

REFERENCES USED:

-Bain J, Plater L, Elliott M, Shpiro N, Hastie CJ, McLauchlan H, Klevernic I, Arthur JS, Alessi DR, Cohen P. The selectivity of protein kinase inhibitors: a further update. *Biochem J.* 2007;408(3):297-315. doi: 10.1042/BJ20070797. PMID: 17850214; PMCID: PMC2267365.

-Logie L, Ruiz-Alcaraz AJ, Keane M, Woods YL, Bain J, Marquez R, Alessi DR, Sutherland C. Characterization of a protein kinase B inhibitor in vitro and in insulin-treated liver cells. *Diabetes.* 2007;56(9):2218-27. doi: 10.2337/db07-0343. PMID: 17563061.

-Dai, S., Gocher, A., Euscher, L. and Edelman A. Serum Starvation Induces a Rapid Increase of Akt Phosphorylation in Ovarian Cancer Cells Volume30, IssueS1 Experimental Biology 2016 Meeting Abstracts, Pages 714.9-714.9 https://doi.org/10.1096/fasebj.30.1_supplement.714.9

-Gao, M., Liang, J., Lu, Y. et al. Site-specific activation of AKT protects cells from death induced by glucose deprivation. *Oncogene* 33, 745–755 (2014). <https://doi.org/10.1038/onc.2013.2>

-Gocher AM, Azabdaftari G, Euscher LM, Dai S, Karacosta LG, Franke TF, Edelman AM. Akt activation by Ca²⁺/calmodulin-dependent protein kinase kinase 2 (CaMKK2) in ovarian cancer cells. *J Biol Chem.* 2017 25;292(34):14188-14204. doi: 10.1074/jbc.M117.778464. PMID: 28634229; PMCID: PMC5572912.

-Maurer U, Preiss F, Brauns-Schubert P, Schlicher L, Charvet C. GSK-3 - at the crossroads of cell death and survival. *J Cell Sci.* 2014;127(Pt 7):1369-78. doi: 10.1242/jcs.138057. PMID: 24687186.

-Moore SF, van den Bosch MT, Hunter RW, Sakamoto K, Poole AW, Hers I. Dual regulation of glycogen synthase kinase 3 (GSK3) α/β by protein kinase C (PKC) α and Akt promotes thrombin-mediated integrin α IIb β 3 activation and granule secretion in platelets. *J Biol Chem.* 2013; 288(6):3918-28. doi: 10.1074/jbc.M112.429936.. PMID: 23239877; PMCID: PMC3567645.

-Manning BD, Toker A. AKT/PKB Signaling: Navigating the Network. *Cell.* 2017;169(3):381-405. doi: 10.1016/j.cell.2017.04.001. PMID: 28431241; PMCID: PMC5546324.

REVIEWERS' COMMENTS

Reviewer #3 (Remarks to the Author):

The issues were well addressed by the authors. I do not have further comments.

NCOMMS-20-15236-B

POINT BY POINT RESPONSES TO REVIEWERS

Reviewer #3 (Remarks to the Author):

The issues were well addressed by the authors. I do not have further comments.

We appreciate the positive comments of the reviewer and his/her suggestions to improve the manuscript